# TRPV4-Rho GTPase complex structures reveal mechanisms of gating and disease

Do Hoon Kwon [1,5], Feng Zhang[1,5], Brett A. McCray [2], Shasha Feng [3], Meha Kumar [2], Jeremy M. Sullivan[2], Wonpil Im [3], Charlotte J. Sumner [2,4] & Seok-Yong Lee [1] ✉

Crosstalk between ion channels and small GTPases is critical during homeostasis and disease, but little is known about the structural underpinnings of these interactions. TRPV4 is a polymodal, calcium-permeable cation channel that has emerged as a potential therapeutic target in multiple conditions. Gain-of-function mutations also cause hereditary neuromuscular disease. Here, we present cryo-EM structures of human TRPV4 in complex with RhoA in the ligand-free, antagonist-bound closed, and agonist-bound open states. These structures reveal the mechanism of ligand-dependent TRPV4 gating. Channel activation is associated with rigid-body rotation of the intracellular ankyrin repeat domain, but state-dependent interaction with membrane-anchored RhoA constrains this movement. Notably, many residues at the TRPV4-RhoA interface are mutated in disease and perturbing this interface by introducing mutations into either TRPV4 or RhoA increases TRPV4 channel activity. Together, these results suggest that RhoA serves as an auxiliary subunit for TRPV4, regulating TRPV4-mediated calcium homeostasis and disruption of TRPV4-RhoA interactions can lead to TRPV4-related neuromuscular disease. These insights will help facilitate TRPV4 therapeutics development.

Although typically considered part of distinct signaling pathways, functional interactions between ion channels and small GTPases have been reported for over two decades and control a range of fundamental processes including cell migration, tumor vascularization, smooth muscle contraction, and mechanosensation, among others[1–6]. Insight into how this dynamic crosstalk is achieved is limited by the absence of resolved channel-GTPase complex structures and structure-guided functional studies.

TRPV4, expressed in the plasma membrane of a wide range of cell types, is a polymodal ion channel whose gating is controlled by multiple endogenous lipids, and exogenous stimuli including synthetic ligands, cell swelling, shear stress, moderate heat, and ultraviolet light[7–12]. TRPV4 mediates calcium-dependent regulation of osmolarity, bone homeostasis, pain, itch, adipose thermogenesis, inflammation,

pulmonary and renal function, integrity of skin and vascular barriers, glial function, joint function, and hippocampal neural function[13–24]. TRPV4 channel activation is linked to numerous disease states including pulmonary edema and cancer metastasis, amongst others[25–28]. Furthermore, gain-of-function missense mutations cause TRPV4 channelopathies, which are grouped into autosomal dominant neuromuscular disorders (Charcot-Marie-Tooth disease type 2C and distal spinal muscular atrophies) and skeletal disorders (skeletal dysplasias and osteoarthropathy)[29–34]. Notably, while skeletal dysplasia mutations are distributed throughout the TRPV4 channel, neuromuscular disease-causing mutations (referred to hereafter as neuropathy mutations) are primarily localized to a confined region of the N-terminal cytoplasmic domain. We previously showed that the cytoskeleton remodeling small GTPase RhoA interacts with TRPV4[35],

[1]Department of Biochemistry, Duke University School of Medicine, Durham, NC 27710, USA. [2]Department of Neurology, Johns Hopkins University School of Medicine, Baltimore, MD 21205, USA. [3]Departments of Biological Sciences, Chemistry, and Bioengineering, Lehigh University, Bethlehem, PA 18015, USA. [4]Department of Neuroscience, Johns Hopkins University School of Medicine, Baltimore, MD 21205, USA. [5]These authors contributed equally: Do Hoon Kwon, Feng Zhang. ✉e-mail: seok-yong.lee@duke.edu

but this interaction appears to be perturbed by neuropathy mutations resulting in increased TRPV4 channel activity, cytoskeletal remodeling, and cell process retraction[35]. Overexpression of RhoA suppresses wild type (WT) TRPV4 channel-mediated calcium influx in cultured mouse motor neuron–neuroblastoma fusion (MN-1) cells in response to hypotonicity, demonstrating its ability to modulate TRPV4 function (Fig. 1a, b), and this effect occurs independent of changes in TRPV4 expression at the plasma membrane[35].

TRPV4 channel inhibition is a promising therapeutic strategy for multiple diseases and conditions[16,36]. Administration of TRPV4 antagonists improves outcomes in animal models of pulmonary edema, blood-retinal and blood-brain barrier breakdown, and peripheral neuropathy[37–39] and the orally bioavailable TRPV4 antagonist GSK2798745 (Fig. 1c) has been safe in clinical trials of pulmonary edema, chronic cough, and diabetic macular edema (NCT02119260, NCT02497937, NCT03372603, and NCT04292912)[36,37,40]. Understanding the structural bases of ligand-dependent TRPV4 gating as well as channel modulation by RhoA will enhance drug design for TRPV4-dependent diseases. To date, the only published structure of TRPV4 is from *Xenopus tropicalis* in its ligand-free, non-conducting state[41], but the atypical non-domain-swapped tetrameric channel arrangement calls into question its physiological relevance[42]. Therefore, little is known about the agonist- and antagonist-binding sites within TRPV4 and how they exert effects on channel gating.

Here we used cryo-electron microscopy (cryo-EM), electrophysiology, MD-simulation, and cell-based assays to investigate the regulation of TRPV4 channel gating by ligands and RhoA. We report four high-resolution cryo-EM structures of human TRPV4 in complex with RhoA in the ligand-free, antagonist-bound closed, and agonist-bound open states. We first identify the specific location of the ligand binding sites and elucidate how ligand binding controls channel gating. We then analyze its interaction with RhoA to elucidate the mechanism of TRPV4 inhibition by RhoA, as well as the impact of human neuropathy mutations on TRPV4 gating.

## Results

### Structures of human TRPV4 in complex with Rho

We expressed full-length WT human TRPV4 in HEK293S GnTI⁻ cells. The protein was extracted and purified in detergent and the structure determined using single-particle cryo-EM (Fig. 1d–f). Surprisingly, although we overexpressed TRPV4 only, an additional protein density associated with each cytoplasmic domain of the tetrameric TRPV4 channel was resolved in the final three-dimensional (3D) reconstruction (Fig. 1d). The published RhoA crystal structure (PDB: 1FTN) fits reasonably well into this density, suggesting that endogenous Rho GTPase was copurified with overexpressed TRPV4. The presence of RhoA in the final purified TRPV4 samples was confirmed by western blot using a RhoA-specific monoclonal antibody (Supplementary Fig. 1a). Given the high sequence homology of the Rho isoforms RhoA, RhoB, and RhoC, we also tested RhoB- and RhoC-specific antibodies and found that RhoB and RhoC also copurified with TRPV4 (Supplementary Fig. 1a). Although RhoA, RhoB, and RhoC share a high degree

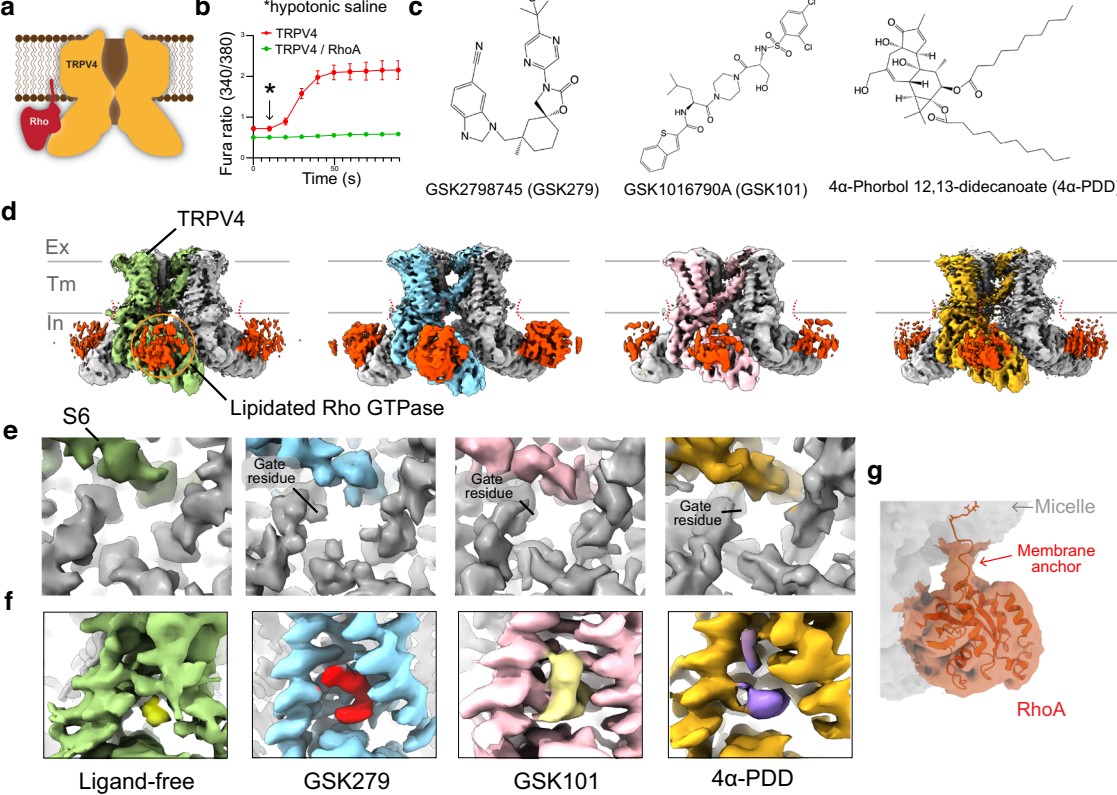

**Fig. 1 | Functional characterization and structure determination of the human TRPV4-Rho GTPase complex. a** Schematic drawing of the functional TRPV4-Rho GTPase interaction. **b** Averaged calcium imaging traces before and after hypotonic stimulation, denoted by the arrow. Expression of wild type human TRPV4 alone causes elevated baseline and stimulated calcium influx relative to the co-expression of human TRPV4 and RhoA. Data are presented as means ± SEM, *n* = 11 wells per condition, with 20–40 transfected cells per well. **c** Chemical structures of GSK1016790A (GSK101), GSK2798745 (GSK279), and 4α-phorbol 12,13-didecanoate (4α-PDD). **d** Cryo-EM structures of the TRPV4-Rho GTPase complex in the ligand-free, GSK279-bound closed, GSK101-bound open, and 4α-PDD-bound states, as indicated. Map thresholding: 0.24 (green), 0.25 (cyan), 0.25 (pink), 0.26 (gold). **e** Close-up view at the S6 gate of 3D reconstructions from (**d**) viewed from the intracellular side, at thresholding 0.19, 0.25, 0.25, and 0.21, respectively. **f** Close-up view at the ligand binding site of 3D reconstructions from (**d**) at thresholding 0.25, 0.36, 0.33, and 0.2, respectively. **g** Cryo-EM density of RhoA and its prenylated tail of the GSK279-TRPV4-RhoA reconstruction, at thresholding 0.09. Source data for (**b**) are provided as a Source Data file.

of sequence homology, the focused cryo-EM map, the docked RhoA model, and the sequence alignment suggest that the final cryo-EM reconstruction is more consistent with RhoA (Supplementary Fig. 2a, b). This suggests that RhoA may be the major Rho GTPase bound to TRPV4 in the final 3D reconstruction although we cannot exclude the possibility that a significant portion of TRPV4 is also bound to RhoB and/or RhoC. For this manuscript, we tentatively assign RhoA as the primary candidate based on the cryo-EM map and the previous identification of RhoA as a TRPV4 binding partner through an unbiased screen[35]. The C1-symmetric 3D reconstruction showed RhoA occupancy at all four subunits of the TRPV4 homotetramer (Supplementary Fig. 1b). We define this ligand-free state of the TRPV4-RhoA complex structure as the apo state. Three additional complex structures were determined in the presence of distinct ligands (Fig. 1c–f): the antagonist GSK2798745-bound TRPV4-RhoA complex in the closed state, the agonist GSK1016790A-bound TRPV4-RhoA in the open state, and the agonist 4α-phorbol 12,13-didecanoate-bound TRPV4-RhoA in the putative open state (ligands are abbreviated as GSK279, GSK101, and 4α−PDD, respectively). The 3D reconstructions of the four structures were resolved to 3.30 to 3.75 Å resolutions (Supplementary Fig. 1c–h). Particle subtraction followed by focused 3D classification of the transmembrane region and RhoA-bound cytoplasmic domain were performed, resulting in improved map qualities for these regions for model building (Supplementary Fig. 3). Although the EM density for the pore domain in the ligand-free state and the 4α-PDD-bound structures was resolved sub-optimally, the high-quality reconstructions for the GSK279-bound closed state and the GSK101-bound open state enabled us to unambiguously model the register and assign the gate residues (Supplementary Fig. 4). Notably, we observed EM density of the RhoA C-terminus extending to the detergent micelle that surrounds the TRPV4 transmembrane region, suggesting that the prenylated C-terminus of RhoA anchors to the inner leaflet of the membrane bilayer and facilitates its association with TRPV4 in the cellular context (Fig. 1g).

Human TRPV4 adopts a two-layered homotetrameric architecture where RhoA attaches to the bottom layer. The top layer, or the transmembrane region, comprises the voltage-sensor-like domain (VSLD) and the pore domain. The VSLD is formed by transmembrane helices S1 to S4, while the pore domain contains the S4–S5 junction, S5, the pore helix (PH), the selectivity filter (SF), the pore loop, the pore-lining helix S6, and the TRP domain (Supplementary Fig. 5a, b). The bottom layer is composed of the cytosolic N-terminal ankyrin repeat domain (ARD; comprising six ankyrin repeats, ARs) and a coupling domain (CD), including a helix-turn-helix motif ($HTH_{CD}$) and β-sheets ($β_{CD}$), and a C-terminal domain (CTD) (Supplementary Fig. 5a). RhoA appears to act as an auxiliary subunit, interacting with the ARD through three loops connecting AR2-AR3, AR3-AR4, AR4-AR5, and with the membrane bilayer through the prenylated C-terminal tail. The apparent stoichiometry of RhoA and each TRPV4 subunit is 1:1 based on the C1 reconstruction (Supplementary Fig. 1b). The human tetrameric TRPV4-RhoA signaling complex structure exhibits a canonical domain-swap tetrameric arrangement where the VSLD of TRPV4 from one subunit interacts with the pore domain from the neighboring subunit (Supplementary Fig. 5b–f). The fold and quaternary structure of full-length human TRPV4 are in stark contrast with the published cryo-EM structure of truncated frog TRPV4, whose pore domains are not swapped, resulting in the Cα RMSD of the pore ~7.4 Å (Supplementary Fig. 5e)[41]. Because the domain-swapped architecture is the well-accepted architecture of the TRP channel superfamily[42], our TRPV4-RhoA structures represent physiologically relevant conformations.

## Agonist- and antagonist-dependent TRPV4 gating

In our ligand-bound 3D cryo-EM reconstructions, we identified strong and unambiguous EM densities corresponding to GSK279 (antagonist), GSK101 (agonist), and 4α-PDD (agonist) located at a shared site

within the cavity between the VSLD and the TRP domain (termed here the VSLD cavity; Figs. 1f, 2a). These compounds are stabilized within the VSLD cavity by many aromatic and polar residues. The additional focused-refinement help further improve the cryo-EM map for the transmembrane regions (Supplementary Fig. 3). We only included the GSK279-bound closed state and the GSK101-bound open state for subsequent analyses of ligand binding and gating due to their high-quality cryo-EM maps.

We functionally probed the interactions of GSK101 with TRPV4 by mutating amino acid residues within the GSK101 binding site using patch clamp recording on HEK293 cells expressing WT as well as mutant TRPV4. Mutations Y553A, N474A, and F524A significantly increased $EC_{50}$ values of GSK101 for TRPV4, consistent with observations from the structural analysis (Fig. 2c, d). We then set out to probe the GSK279 interactions with TRPV4. The shared binding region for GSK279 and GSK101 complicates mutagenesis studies to specifically examine GSK279 interactions with TRPV4. Therefore, we utilized an approach to create a 2-aminoethoxydiphenyl borate (2-APB) agonist binding site in TRPV4 distinct from the VSLD cavity. Patapoutian and colleagues previously demonstrated the 2-APB binding site in TRPV3 and created an analogous 2-APB responsive site in TRPV4 by site-directed mutagenesis (N456H/W737R, denoted as $TRPV4^{DM}$) that enabled $TRPV4^{DM}$ activation by 2-APB[43] (Supplementary Fig. 6a). As the 2-APB binding site in TRPV3 is located at a distance from the VSLD cavity[44], we predicted that 2-APB binding to $TRPV4^{DM}$ would not interfere with either GSK279 or GSK101 binding. To verify that $TRPV4^{DM}$ does not disrupt TRPV4 ion channel function or GSK101 binding, we demonstrated that 1) $TRPV4^{DM}$ can be activated by either osmotic stimuli or GSK101 to a similar extent as WT TRPV4 (Supplementary Fig. 6b, c) and 2) GSK101 binding site mutants Y553A, D743A, and F524A introduced onto the background of $TRPV4^{DM}$ suppressed TRPV4 activation by GSK101 relative to that by 2-APB (Supplementary Fig. 6d), similar to the results from the WT TRPV4 background. We observed that D743A and D546A mutations of $TRPV4^{DM}$ attenuate channel inhibition by GSK279 in our two-electrode voltage-clamp (TEVC) recordings (Fig. 2f, g and Supplementary Fig. 6e, f). Previous studies have suggested that the S2-S3 loop, S3, and S4 are involved in TRPV4 sensitivity to the inhibitor HC067047 and the agonist 4α-PDD[45,46], which is consistent with our findings (Fig. 2). We further examined the ligand binding poses by using unstrained all-atom molecular dynamics (MD) simulations. The GSK101 binding pose remained stable during the simulations, supporting its accuracy. For GSK279 binding poses, we tested two poses (one with the best fit to the cryo-EM density, the other as an alternative binding pose). The original binding pose (pose I) remains stable while the alternative binding pose (pose II) exhibits dynamic movement, providing further support for the pose I (Fig. 2h, i). For 4α−PDD binding poses (Supplementary Fig. 6g–i), we tested three poses, and pose III remains relatively stable while the other two poses are dynamic, supporting that pose III is likely the binding mode of 4α-PDD to TRPV4 (Supplementary Fig. 6i, j). Notably, an aliphatic side chain attached to the 4α-phorbol moiety interacts with the hydrophobic groove of the VSLD in the cryo-EM data, and during the MD simulation this interaction remains stable, indicating the importance of this interaction for TRPV4 binding, consistent with the functional studies on 4α-PDD and different 4α-phorbol esters[47] (Supplementary Fig. 6g–j).

How is it that GSK279 and GSK101 impose opposite effects on TRPV4 gating while binding to a common set of residues in the VSLD cavity? We observe that going from the GSK279-bound closed state to the GSK101-bound open state, the S2-S3 linker (M534-S548) undergoes a loop-to-helix transition, thereby altering the interaction network among the VSLD, TRP domain, and CD (Fig. 3a). In the closed state, D531 (S2), Q550 (S3), D546 (S2-S3), and R594 (C-terminal half of S4; S4b) form a charged H-bond relay within the VSLD, while D743, R746 (TRP domain and Y439 (CD; $HTH_{CD}$) connect the TRP domain and the CD (Fig. 3b,

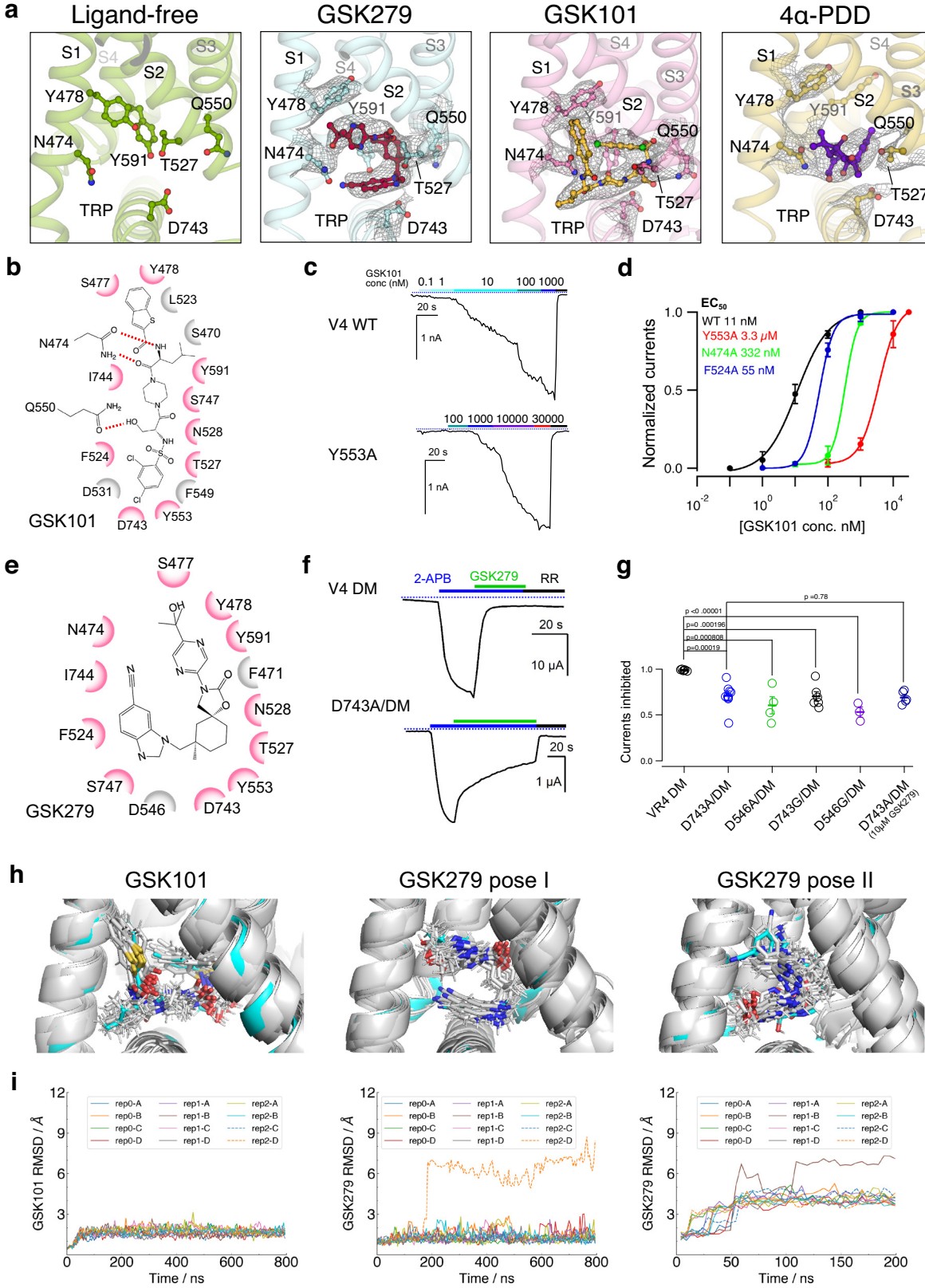

left). These two separate interaction networks appear to be decoupled between the VSLD and the TRP/CD. In contrast, the rearrangement of the S2-S3 linker in the open state couples the VSLD to the TRP domain and the CD. First, the D531-Q550-D546-R594 interactions within the VSLD are broken. As a result, Q550 (S3) interacts with both the hydroxyl group of GSK101 and N528 (S2), while R594 (S4b) forms a new salt-

bridge interaction with D743 (TRP domain), and D531 (S2) interacts with R746 (TRP domain) (Fig. 3b, right), resulting in the ~3 Å swing of the TRP domain and rearrangement of the CD (HTH_CD) (Fig. 3c, d). Therefore, the opposing actions of agonist/antagonist binding to the same cavity in TRPV4 originate from the ligand-dependent coupling or decoupling of the VSLD-TRP-CD subdomains (Fig. 3b).

**Fig. 2 | Antagonist- and agonist-binding in the TRPV4 channel. a** Cryo-EM densities (gray mesh) for ligands (GSK279; red stick, GSK101; gold stick, and 4α−PDD; violet stick) in ligand-free, closed, open, and 4α-PDD-open states. Densities are contoured at 0.23, 0.36, 0.33, and 0.195 thresholding, respectively. Sidechains of key residues are shown in sticks. **b** Ligplot schematics of GSK101-TRPV4 interactions, where residues within 3.8 Å to the ligands are shown. Pink colored residues are involved in both GSK101 and GSK279 bindings. **c** Representative patch clamp recording of wild type TRPV4 and mutant Y553A at −60 mV at increasing concentrations of GSK101, followed by block with ruthenium red (RR, 50 μM), as indicated by the colored horizontal lines. The blue-dotted lines indicate the zero-current level. **d** Mean normalized concentration-response relations for GSK101. Data are shown as mean ± SEM. (*n* = 3–5 oocytes). The continuous curves are fits to the Hill equation with EC₅₀ as indicated in the figure. **e** Ligplot schematics of GSK279-TRPV4 interactions. **f** Representative two-electrode voltage-clamp recording of TRPV4 mutant (TRPV4^DM), and additional mutants made with

the background TRPV4^DM as indicated in the figure. **g** Summary of inhibition by GSK279 relative to current from saturating 2-APB (2 mM) at room temperature. Values for individual oocytes are shown as open circles with mean ± SEM shown (The *n* values are 6, 9, 4, 6, 3 oocytes, respectively. For D743A with 10 μM GSK279 inhibition, the *n* = 5 oocytes.). *P* values are calculated by two-tailed Student's *t* test as indicated in the figure. **h** Ligand-binding conformational ensemble from 12 replicas of GSK101 (left), GSK279 pose I (middle) and GSK279 pose II (right). **i** Ligand RMSD values of GSK101 show stable ligand binding with an average RMSD of 1.65 Å. Each trajectory represents a subunit (A/B/C/D) in one of the three replicas (left). Ligand RMSD values of GSK279 pose I show stable ligand binding with an average RMSD of 1.28 Å, except for one outliner ligand, rep2-D, which stumbles out of the pocket (middle). Ligand RMSD values of GSK279 pose II show large deviations from the initial configuration with an average RMSD of 4.33 Å (right). Source data for (**c**, **d**, **f**, and **g**) are provided as a Source Data file.

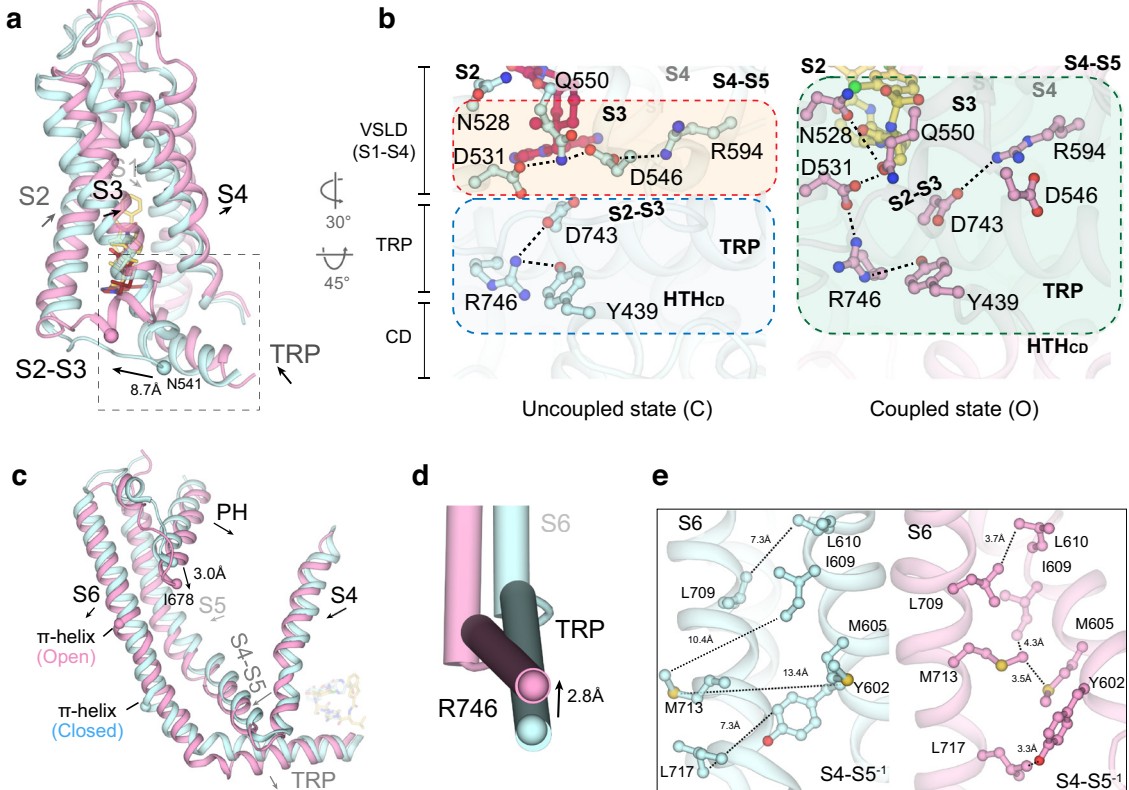

**Fig. 3 | Ligand-dependent conformational changes of TRPV4. a** Comparison of conformational changes in the VSLD of GSK279-TRPV4-RhoA (cyan) and GSK101-TRPV4-RhoA (pink). Ligands are shown as sticks. Arrows indicate helix movements and rotations. **b** Comparison of coupling networks at the VSLD, CD and TRP domain in closed and open states. Dashed lines indicate hydrogen bonds and salt bridges. **c** Comparison of conformational changes at the pore domain of GSK279-TRPV4-RhoA (cyan) and GSK101-TRPV4-RhoA (pink). Ligands are shown in sticks. Arrows

indicate helix movement with distances indicated between reference points (as spheres). **d** Close-up view of S6b and the TRP helix. Arrows indicate helical movements with distances indicated between reference points (as spheres). **e** Side-by-side comparison of the closed (cyan) and open states (pink) at inter-subunit interfaces. Sidechains are shown as sticks. Dashed lines indicate the distances between corresponding residues.

These conformational changes, which are associated with channel activation, propagate from the ligand-binding site to the pore domain. The S4-S5 linker, S5, and PH rotate toward the central ion conduction pathway, while S6 undergoes changes in the π-helix position (from M718 to V708) and rotation at the C-terminus (C-terminal half of S6 [S6b]; residues N712-G719) (Fig. 3c−e).

### Pore conformation changes during TRPV4 gating
We observe extensive conformational changes in the TRPV4 pore domain during ligand-dependent gating. The GSK279-bound closed state adopts a wide-set SF (I678-G679-M680) and the narrowest constriction point of the S6 gate is at M718 (Fig. 4a−c). The diagonal

distance between G679 backbone carbonyls in the SF (12.2 Å) is too far to directly coordinate cations, while that between the M718 sidechains (4.9 Å) is too narrow for ion conduction. However, in the GSK101-bound open state, the SF and the PHs move closer to enable cation coordination (the diagonal distance of G679 backbone carbonyl is 7.0 Å), which is consistent with previous mutagenesis studies[48]. Meanwhile the pore-lining S6 helices rotate ~90°, thus changing the gate position (from M718 to I715) and the gate opening (the diagonal distance at I715 is 8.3 Å) (Fig. 4c, d). This S6 rotation between the two states is due to the shift of the π-helix position on S6 (Fig. 4a), resulting in a local secondary structure conversion between α- and π-helix along S6b, which is unique amongst the TRP

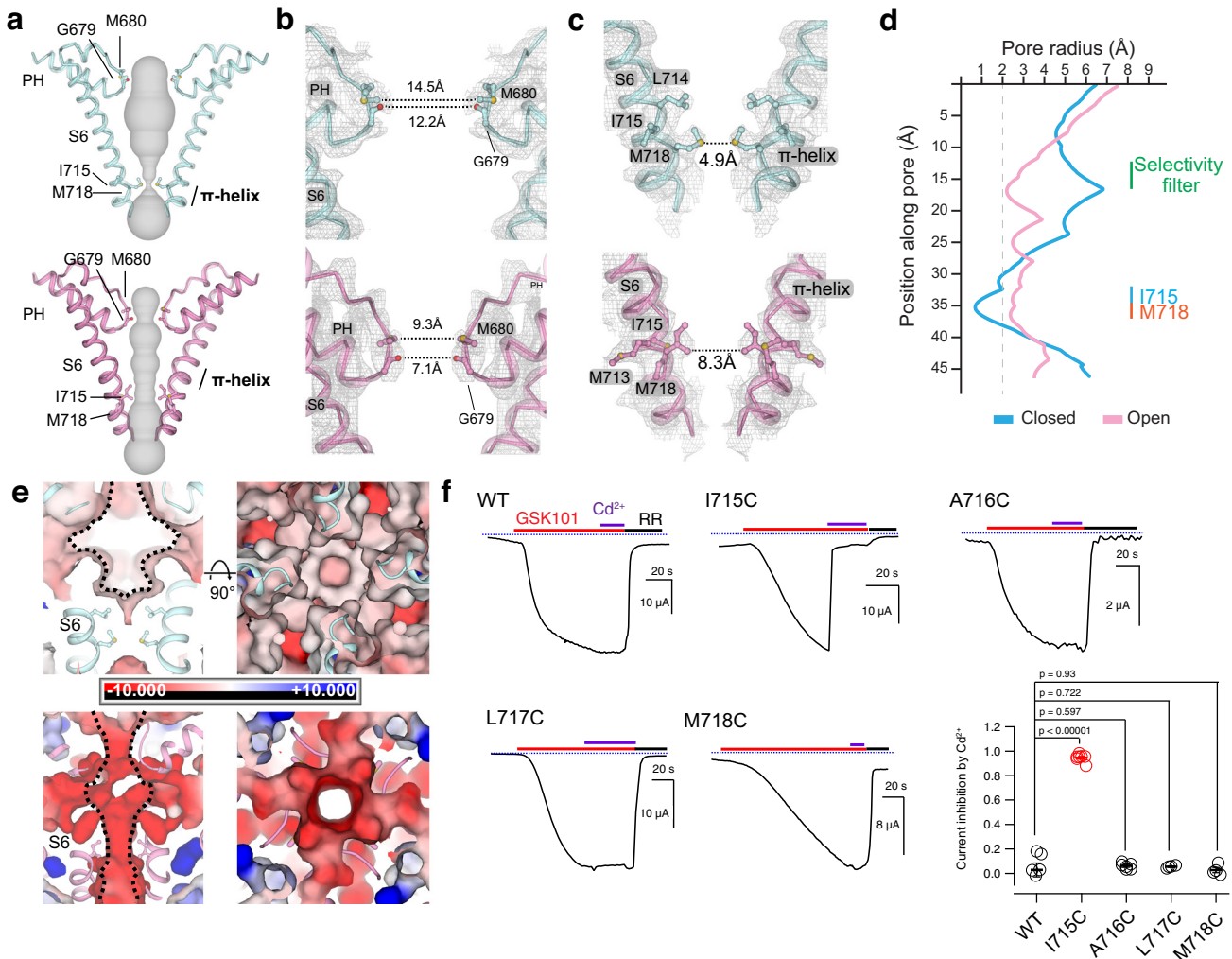

**Fig. 4 | Structural changes in the S6 gate and the SF of the pore during channel gating. a** Ion permeation pathway in the closed and open states shown as gray surfaces. S6 helices from two protomers are shown in cartoon. Only two subunits are shown for clarity. Gate and selectivity filter residues shown as sticks. **b**, **c** Close-up views of the SF region (**b**) and the S6 gate (**c**) for the closed and open states. The dotted lines indicate diagonal distances between gating residues of opposite protomers. Gray mesh indicates cryo-EM densities of TRPV4-focused maps contoured at 0.3 (top) and 0.35 (bottom) thresholding, respectively. **d** Pore radii calculated using the HOLE program in Coot for representative TRPV4 structures as color-coded. The minimal radius for a hydrophobic gate to be open is considered 2.0 Å. Residues corresponding to the SF (M680 and G679) and the S6 gate (I715 and M718) are denoted. **e** APBS surface electrostatics of the pore in the closed and open states as viewed from the membrane plane (right) and from the extracellular side (left). S6 helices and SF region are shown in cartoon and gating residues as sticks. **f** Representative time-course recording of WT TRPV4 and mutants. Currents elicited by 5 μM GSK101 and co-application with 10 μM $Cd^{2+}$ followed by 20 μM ruthenium red (RR) as indicated by colored horizontal lines. The voltage was ramped from −60 mV to +60 mV in 300 ms every 2 seconds. The currents at −60 mV were used for the plot. Dotted blue lines indicate zero-current level. Right panel, summary of current inhibition by 10 μM $Cd^{2+}$ relative to 5 μM GSK101-induced currents. Values for individual oocytes are shown as open circles with mean ± SEM (WT n = 6, I715C n = 7, A716C n = 6, L717C n = 4, M718C n = 4), P values are calculated by two-tailed Student's t test as indicated in the figure. Source data for (**f**) are provided as a Source Data file.

channels[49]. The conformational changes of S6 are facilitated by rearrangement in the interfacial contacts with the neighboring S4-S5 linker (Fig. 3e). Interestingly, when compared to the closed state, we found that the pore cavity volume is reduced, and the surface electrostatic potential becomes substantially more negative in the open state (Fig. 4e). The constriction of the SF and the PH, the expansion of the S6 gate, and the surface electrostatic changes of the cation permeation pathway during TRPV4 activation are unique amongst TRPV channels[42,50]. However, a similar motion has been observed in recent studies of agonist- and lipid-dependent TRPM8 activation[51]. Taken together, we posit that these unusual changes in the pore help form the ion permeation path, similar to the canonical tetrameric cation channel[42].

To validate the S6 gate position observed in the open-state structure, we performed $Cd^{2+}$-dependent blocking of TRPV4.

Extracellular application of $Cd^{2+}$ does not affect the opening of WT TRPV4 by GSK101 (Fig. 4f). Among cysteine mutants near the S6 gate (I715C, A716C, L717C, and M718C), only I715C showed substantial inhibition of GSK101-elicited current by extracellular $Cd^{2+}$, supporting the observation that I715 faces the ion permeation pathway in the TRPV4 open state (Fig. 4f).

## Neuropathy mutations disrupt RhoA binding to TRPV4

At the core of RhoA is a highly conserved G domain comprising a six-stranded β-sheet (β1-β6), six helices (α1-α6), and a C-terminal variable region (Fig. 5a and Supplementary Fig. 2a, b). Switch I and II regions within the G domain adopt distinct GDP- or GTP-dependent conformations, where the latter favors effector protein binding[52].

We performed focused-refinement on the RhoA and ARD region in the 3D reconstructions of the GSK279-bound, ligand-free, and

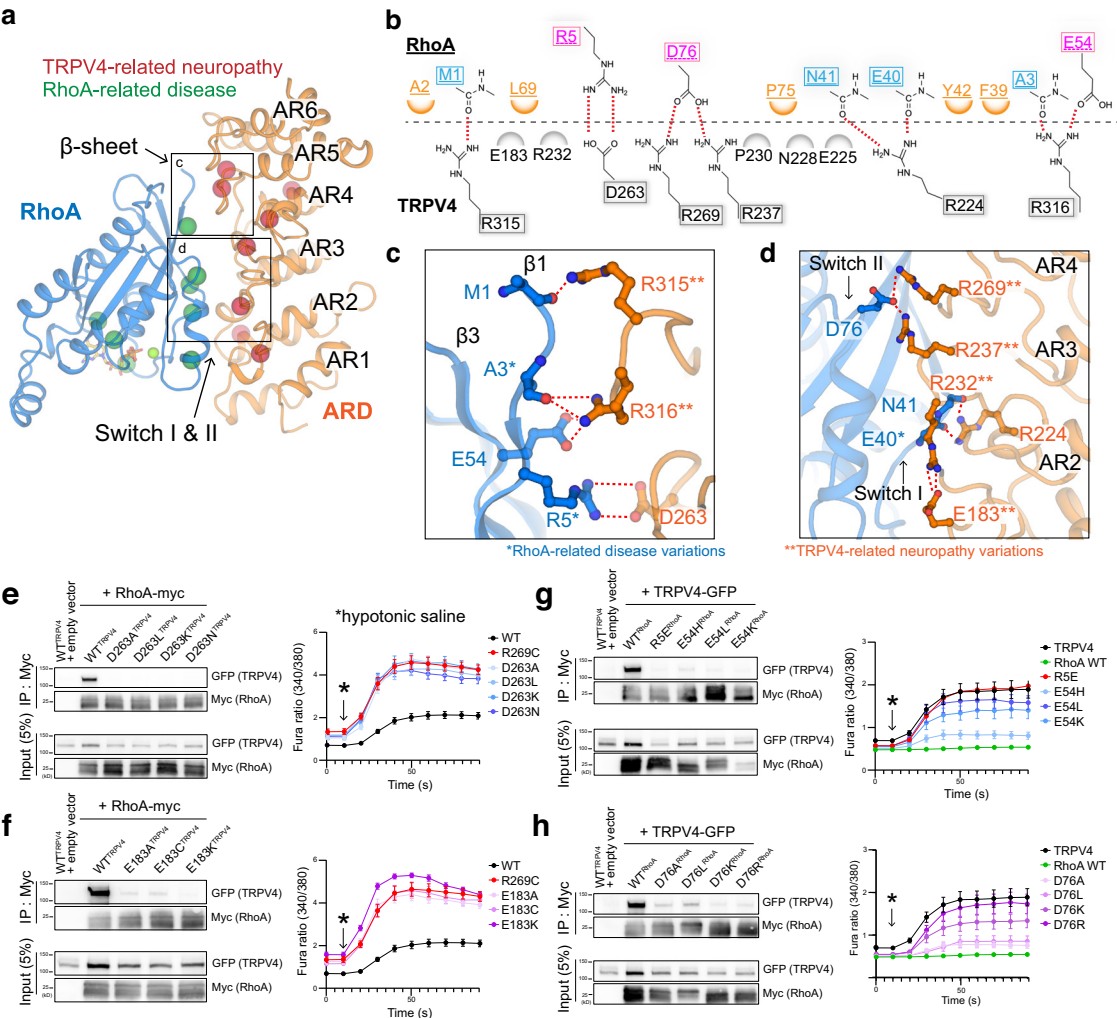

**Fig. 5 | Interaction between TRPV4 ARD and RhoA. a** Overall interaction interface of TRPV4 ARD and RhoA. Disease-causing mutations mapped onto the TRPV4-RhoA interface. Disease-causing mutations of TRPV4 and RhoA are shown as red and green spheres, respectively. **b** DIMplot[98] schematics of TRPV4 ARD and Rho GTPase interactions, with critical positions labeled. Pink-colored residues indicate interactions via sidechains. Blue-colored residues indicate backbone-sidechain interactions. Orange-colored residues indicate hydrophobic interactions.
**c, d** Detailed TRPV4 ARD-RhoA interactions within the β1-β3 region (**c**) and switch region (**d**). RhoA residues are colored blue, TRPV4 residues are colored orange. * disease-causing mutations in RhoA and **neuropathy-causing mutations in human TRPV4. The red dashed lines indicate salt bridge interactions. **e**–**h** (Left panels) Co-

immunoprecipitation of HEK293T cells transfected with TRPV4-GFP (**e** E183A/C/K; **f** D263A/L/K/N) and RhoA-Myc (**g** R5E, E54L/H/K; **h** D76A/L/K/R) demonstrates that mutations at the TRPV4-RhoA interface reduces their interaction. (Right panels) Averaged ratiometric calcium plots from ratiometric calcium imaging experiments. MN-1 cells were transfected with GFP-tagged TRPV4 plasmids only (**e, f**) and GFP-tagged TRPV4 and RhoA-Myc plasmids (**g, h**) and loaded with Fura-2 AM calcium indicator. Baseline and hypotonic saline-stimulated calcium responses were then measured over time. $N = 9$ wells per condition, with 20–40 transfected cells per well. Data are shown as mean ± SEM. Source data for (**e**–**h**) are provided as a Source Data file.

GSK101-bound data. Although the EM density for the ARDs in all three states exhibited high quality, that for RhoA was most well resolved in the GSK279-bound closed state, which we used to unambiguously model RhoA and map its interaction with TRPV4 (Supplementary Figs. 2b, 4a, b). We resolved GDP within the RhoA-TRPV4 signaling complex in the GSK279-bound closed state, and by including GTPγS before freezing grids we observed density consistent with GTPγS in the open state (Supplementary Fig. 7a, b). We did not, however, observe substantial structural differences of RhoA between the GDP- and GTPγS-bound forms in complex with the closed and open states of TRPV4, respectively (Supplementary Fig. 7c). We used the model from the GSK279-bound closed state for our analysis of RhoA-TRPV4 interactions. Notably, while the switch I in RhoA resembles the GDP-bound RhoA structure, the switch II is distinct from either the GDP- or the GTP-bound RhoA structure (PDB IFTN and 1A2B; Supplementary Fig. 7d). The interfacial contact that associates TRPV4 and RhoA is

mediated principally by electrostatic interactions between β1 and β3, switch I, and switch II in RhoA and AR2-AR5 in TRPV4 (Fig. 5a, b). Sequence comparisons indicate that the interfacial residues are unique to the TRPV4 ARD across all TRPV channels, and to Rho isoforms amongst other small GTPases, suggesting that the observed RhoA binding mode is specific to TRPV4 (Supplementary Fig. 8)[35].

Strikingly, most residues mutated in TRPV4-mediated neuropathy (R237, R269, R315, and R316) are clustered at the interface with RhoA (Fig. 5a). R232, a reported TRPV4-neuropathy mutation site, does not directly participate in the interfacial contact but does form an intra-subunit salt-bridge with another disease-causing residue E183, which is likely important for tuning electrostatics and the local conformation of the RhoA-binding surface of the ARD (Supplementary Fig. 7e). Therefore, disease-causing mutations in TRPV4 likely weaken the interactions with RhoA[35]. Notably, we also found that several cancer-related mutation sites in RhoA, such as A3, R5, and E54[53–55], are located at the

interface with the TRPV4 ARD (Fig. 5c, d), providing evidence for an interplay between TRPV4 and RhoA in cancer[25,26,56], although these RhoA mutations may also impact effector binding more broadly.

To probe the interactions between TRPV4 and RhoA, we first performed co-immunoprecipitation (co-IP) of GFP-tagged WT or mutant TRPV4 with Myc-tagged WT or mutant RhoA overexpressed in HEK293T cells[35]. We previously showed that neuropathy-causing mutations (R232C, R237L, R269C, R315W) in TRPV4 disrupt the interaction with RhoA, consistent with our structures (Fig. 5c, d). We further mutated residues in RhoA (R5, E54, D76) and TRPV4 (E183 and D263) that form the RhoA-TRPV4 interface. All mutants tested (R5E$^{RhoA}$, E54H/L/K$^{RhoA}$, D76A/L/K/R$^{RhoA}$, E183A/C/K$^{TRPV4}$, and D263A/L/K/N$^{TRPV4}$) substantially decreased the amount of immunoprecipitated partner proteins (TRPV4 and RhoA) (Fig. 5e–h). We then tested the effects of these mutations on TRPV4 function using ratiometric calcium imaging. With expression of TRPV4 mutants that fail to interact with RhoA (TRPV4 E183A/C/K or D263A/L/K/N), we found increased basal and hypotonic saline-induced calcium influxes, similar to the neuropathy mutant R269C[35]. This suggests that disruption of interaction with endogenous RhoA leads to increased ion channel activity in the mutants. To directly test this possibility, we took advantage of the fact that the RhoA inhibitor exoenzyme C3 transferase of *C. botulinum* binds to RhoA within the TRPV4-RhoA interface[57]. As predicted, treatment of cells with C3 transferase prior to co-immunoprecipitation strongly disrupted TRPV4-RhoA interaction (Supplementary Fig. 9a, b). In addition, treatment of MN-1 cells with C3 transferase led to a marked increase in hypotonic saline-induced calcium influx (Supplementary Fig. 9c). These results suggest that disruption of RhoA interaction alone results in increased TRPV4 ion channel function. We then tested the effect of RhoA mutations on TRPV4 channel activity, in experiments in which we overexpressed both TRPV4 and RhoA. In this paradigm, we previously showed that over-expression of RhoA suppresses TRPV4 ion channel activity in response to hypotonic saline or GSK101[35], perhaps due to the inability of endogenous RhoA to fully inhibit over-expressed TRPV4. Whereas expression of WT RhoA strongly suppressed both basal and hypotonic saline-induced Ca$^{2+}$ influx (Figs. 1b, and 5g, h) consistent with prior results, RhoA mutants demonstrated reduced suppression of TRPV4 channel activity (Fig. 5g, h). Notably, there was a correlation between the degree of suppression of TRPV4 activity and the interaction strength between the RhoA mutants and TRPV4 (Fig. 5g, h, and Supplementary Fig. 9d), with the mutants with the highest residual TRPV4 binding (D76A/L and E54H) showing the strongest suppression of TRPV4 channel activity. These data suggest that TRPV4-RhoA interaction strength correlates with TRPV4 channel activity, and that disruption of this interaction underlies the gain of function due to neuropathy mutations within the ARD. This indicates that balancing the two different neurophysiological signaling pathways (calcium signaling and actin cytoskeleton remodeling) may require fine tuning of the TRPV4-RhoA interaction, as was suggested from the previous study[35].

## RhoA-mediated TRPV4 inhibition

Despite only subtle conformational changes in RhoA, the buried surface area between RhoA and TRPV4 in the open state is reduced (~684 Å$^2$) compared to that observed in the closed state structure (~752 Å$^2$), suggesting RhoA-TRPV4 interactions become weaker in the open state (Supplementary Fig. 7f). To compare the relative occupancy and/or dynamics of RhoA bound to TRPV4 in different channel functional states, we low-pass filtered the cryo-EM maps of the closed, ligand-free, and open state structures at the same resolution (4 Å) and applied the same value of B-factor sharpening. Notably, the EM density for RhoA progressively decreases from the closed to the open states (Fig. 6a). Consistent with this observation, further cryo-EM 3D classifications of the closed and the open state reconstructions show that the closed state reconstruction contains a major class with strong RhoA density

while the open state reconstruction contains classes with weak RhoA density (Supplementary Fig. 10). The apparent differential resolution of RhoA density between the closed, ligand-free, and open states led us to hypothesize the state-dependent RhoA binding affinity to TRPV4. Because we include GSK101 before freezing, this data suggests that either bound RhoA becomes more flexible, or its occupancy decreases during GSK101-dependent TRPV4 activation (Fig. 6a). We previously showed that TRPV4 stimulation with hypotonic saline induces RhoA dissociation from the complex[35], but we found no significant dissociation of RhoA in the presence of GSK101 in our co-IP experiments (Supplementary Fig. 9e), suggesting that RhoA becomes more flexible when TRPV4 is activated with GSK101 and is released with osmotic stimuli. These data, taken together, suggest that RhoA binding stabilizes the closed state of TRPV4 and that stimulus-specific TRPV4 activation leads to either RhoA release from or increased flexibility within the complex. Consistent with this idea, the all-atom MD simulations showed that RhoA binding reduced the ARD fluctuation substantially in the GSK279-bound closed state based on the root-mean-square-fluctuation (RMSF) plot (Fig. 6b). Furthermore, the MD simulation showed that bound RhoA is more flexible with GSK101 (Supplementary Fig. 11).

TRPV channels are highly allosterically coupled across their domains, as was previously shown[58], and because we observed progressive changes in RhoA occupancy/dynamics during gating, we attempted to infer the effect of RhoA on TRPV4 gating. When viewed from the cytoplasmic side, TMD-aligned structures reveal that individual ankyrin repeats (ARs) rotate clockwise rather than a concerted rotation of the tetrameric ARD ring (Fig. 6c). In addition to the clockwise rotation each AR in the ARD swings toward the membrane by ~6 Å from the closed to the open state (Fig. 6d). Movements in the ARD propagate as conformational changes in the CD (HTH$_{CD}$), and in turn the TRP domain and the VSLD (Fig. 6e), as essential gating steps in TRPV4 activation (Fig. 2). Taken together, we propose the following model. When membrane-anchored RhoA binds to TRPV4 at the cytoplasmic ARD, it exerts forces on the ARD analogous to a clamp, which suppresses ARD motion associated with TRPV4 activation, resulting in TRPV4 inhibition (Fig. 6f). Our model is consistent with the results from our cryo-EM and computational analyses (Fig. 6a, b and Supplementary Fig. 11).

## Discussion

Here we resolve human TRPV4 structures in complex with the small GTPase RhoA and elucidate structural mechanisms underlying the crosstalk between RhoA and TRPV4. The rigid-body rotation of the TRPV4 ARD domain couples the CD, the TRP domain, and VSLD, leading to TRPV4 opening (Figs. 3b and 6e). Through membrane-anchored RhoA binding, nature has designed a means to control the ARD rearrangement and therefore TRPV4 gating. Membrane anchoring via prenylation plays crucial roles in small GTPase functions[59,60]. In this case, anchoring likely enhances RhoA interactions with TRPV4 ARD by increasing local concentration and imposing geometric constraints. Although our current studies focus on the effect of RhoA on TRPV4 function, conversely, TRPV4 activity can modulate RhoA function[35,61]. Our previous work showed that in the case of activation of TRPV4 by osmotic stimulation, membrane-bound RhoA is released from TRPV4 to allow regulation of cytoskeleton dynamics[61].

While the localization of neuropathy-associated TRPV4 mutations at the TRPV4-RhoA interface suggests that control of this signaling complex is particularly important in the nervous system, TRPV4-RhoA interactions are likely to play a fundamental role in signaling in other disease states as well. Intriguingly, we demonstrated that several cancer-related mutations of RhoA also disrupt the TRPV4-RhoA binding interface. Although these mutations may interrupt RhoA binding with several partners, these data are consistent with prior work demonstrating a role for TRPV4-RhoA in cancer[25,26,56]. TRPV4-RhoA

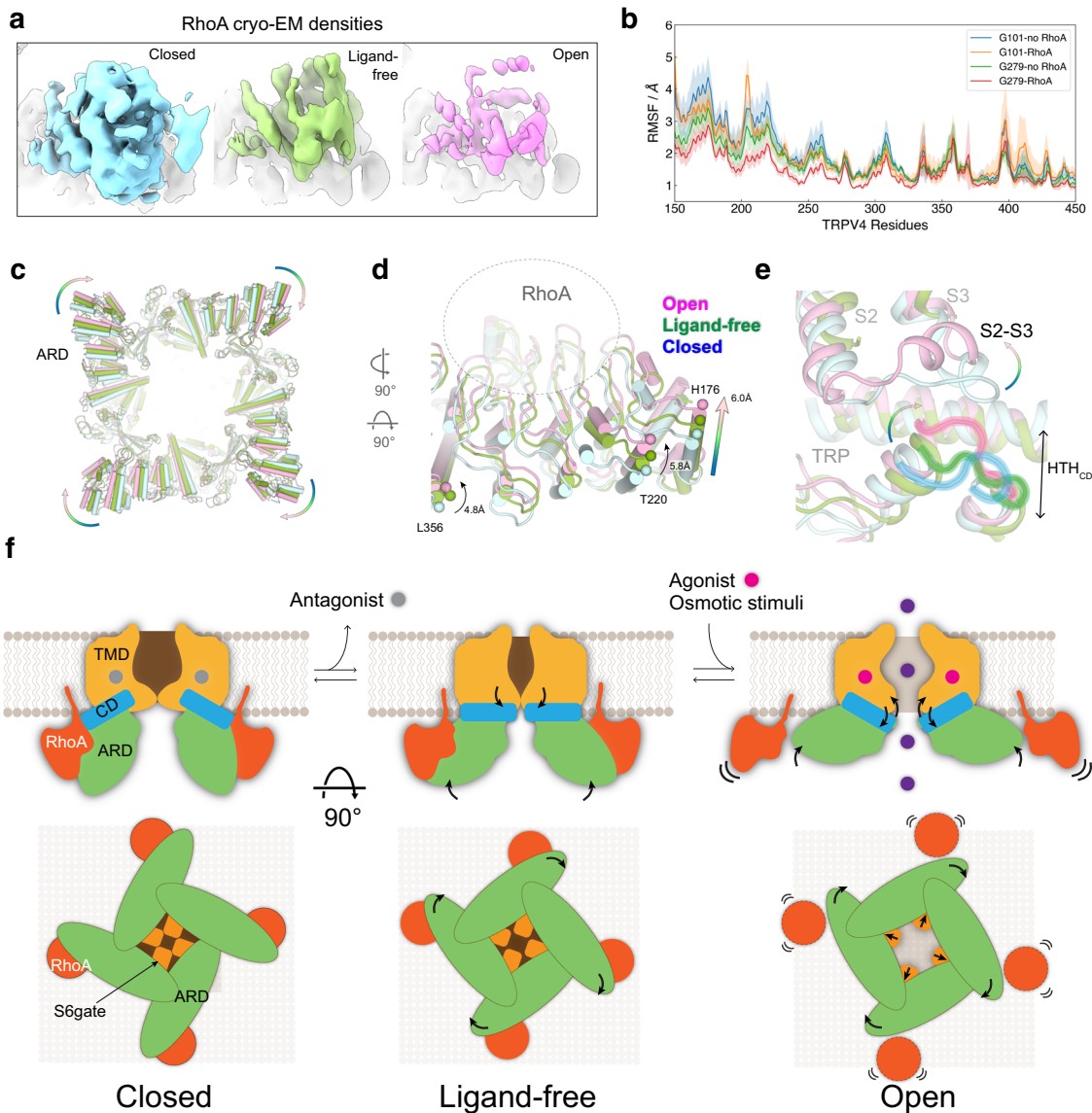

**Fig. 6 | Structural basis of RhoA-dependent gating of TRPV4. a** Side-by-side comparison of cryo-EM densities of RhoA in closed (cyan), ligand-free (green), and open (pink) states at thresholding 0.25. 4 Å low-pass filter and the same B-factor (−50) are applied to all cryo-EM maps. **b** Root-mean-square-fluctuation (RMSF) of residues in the TRPV4 ARD in GSK101-TRPV4, GSK101-TRPV4-RhoA, GSK279-TRPV4, and GSK279-TRPV4-RhoA systems from all-atom MD simulations. In the GSK279-bound state, RhoA binding significantly reduced ARD fluctuation. Shades indicate standard deviations from 12 replicas. **c–e** Comparison of closed, ligand-free, and open structures viewed from the intracellular side (**c**) Close-up view of the ARD (**d**) and coupling domain (**e** HTH_CD, TRP, and S2–S3). Arrows indicate movements of the ARD. ARD/CD rigid-body movement occurs at an individual protomer level. **f** Ligand-dependent channel gating of TRPV4-RhoA. Schematic illustration of the conformational rearrangements in the S6 gate, ARD, and RhoA during TRPV4 gating.

interactions are also likely essential in endothelial (e.g. lung, retinal, intestinal) and epithelial (e.g. skin, lung) cells, where both proteins have been shown to play key roles in barrier maintenance[26,56,62]. Importantly, numerous studies have reported that RhoA and other small GTPases can affect the function of various ion channels, including K+ channels, several other TRP channels, epithelial Na+ channels, and acid-sensing ion channels[2–6]. Further work will be needed to determine whether these other functional channel-GTPase couplings involve direct interactions, and if so, whether similar or distinct structural mechanisms are utilized to those described here.

TRPV4 is a well-established osmosensor[8] and is also reported to respond to shear stress, but the structural mechanisms mediating channel activation by these stimuli have not been determined. Osmosensors can sense either the changes in the extracellular water activity or the resulting changes in cell structure[63]. We speculate that

the wide-set pore domain in the closed state of TRPV4 and its unusual pore conformational changes upon channel activation contribute to channel sensitivity to changes in extracellular fluid activity or shear stress, although we cannot rule out the involvement of other proteins and lipids[17,64]. Moreover, the direct association with plasma membrane-anchored RhoA may enable TRPV4 to detect changes in cellular shape and morphology induced by osmotic shock or mechanical force. Consistent with this speculation, RhoA is activated under cell stretch[65], and the interaction of TRPV4 with actin is essential for cell swelling-induced channel activation[66].

Our studies also deciphered mechanisms of ligand-dependent TRPV4 activation and inhibition. The binding of synthetic antagonists and agonists to a shared site within the VSLD cavity leads to opposing conformational changes of the TRPV4 pore. Our structural analysis elucidated the molecular principle that agonist binding couples,

whereas antagonist binding decouples, the interaction network amongst the VSLD, the TRP domain, and the CD. Recent studies have revealed that lysophosphatidylcholine (LPC) acts as an endogenous agonist, binding to R746 within the TRP domain[67]. It remains to be determined if this endogenous agonist employs an activation mechanism similar to that of synthetic agonists, which could potentially aid in further development of TRPV4 agonists. Importantly, our discoveries of the human TRPV4 structure bound to the clinical candidate antagonist GSK279, as well as the molecular mechanisms of ligand-dependent TRPV4 gating will facilitate the development of drugs targeting TRPV4. Notably, none of the TRPV4 disease mutations[68] overlap with the GSK279 binding site, making it a promising target for future TRPV4 inhibitors. Therefore, with this structure, one can modify GSK279 or find compounds with new scaffolds (via virtual screening) to increase their potency or efficacy.

## Methods

### Protein expression and purification

The *Homo sapiens* full-length TRPV4 (hTRPV4) was cloned into a modified pEG BacMam vector[69] in frame with a FLAG-tag and 10× His-tag at the C terminus. The hTRPV4-RhoA complex was expressed by baculovirus-mediated transduction of human embryonic kidney (HEK) GnTI⁻ suspension cells, cultured in FreeStyle 293 medium (Life Technologies) with 2% (v/v) FBS at 8% $CO_2$. TRPV4 baculovirus was generated following the Bac-to-Bac® Baculovirus Expression System protocol (Life Technologies). Cultures were infected at a cell density of ~1.5 M mL⁻¹ with 1.5–2% (v/v) P3 or P4 baculovirus for TRPV4. After 18 h of shaking incubation at 37 °C, 10 mM sodium butyrate was added, and the growth temperature was lowered to 30 °C to boost protein expression. After 68–72 h, the cells were harvested by centrifugation at 550 x g and resuspended in lysis buffer (20 mM Tris pH 8, 150 mM NaCl, 12 µg ml⁻¹ leupeptin, 12 µg ml⁻¹ pepstatin, 12 µg ml⁻¹ aprotinin, 2 µg ml⁻¹ DNase I, 1 mM PMSF, 10% (v/v) glycerol, 1 mM dithiothreitol (DTT), 1% (w/v) Lauryl Maltose Neopentyl Glycol (LMNG; Anatrace), and 0.1% (w/v) cholesteryl hemisuccinate (CHS; Anatrace). Membranes were solubilized at 4 °C by gentle agitation for 2 h followed by centrifugation at 8000 x g for 30 min to remove insoluble material. The supernatant was subsequently incubated with anti-FLAG M2 resin (Sigma-Aldrich) for 40 min at 4 °C with gentle agitation. The resin was then packed onto a gravity-flow column (Bio-Rad) and washed with 10 column volumes (CV) of high salt wash buffer (20 mM Tris pH 8.0, 500 mM NaCl, 0.1% LMNG, 0.01% CHS, 1 mM DTT, and 5% glycerol) followed by 10 CV of low salt wash buffer 1 (20 mM Tris pH 8.0, 150 mM NaCl, 0.1% LMNG, 0.01% CHS, 1 mM DTT, 5 mM ATP, and 5% glycerol), and then 10 CV of low salt wash buffer 2 (20 mM Tris pH 8.0, 150 mM NaCl, 0.03% LMNG 0.003% CHS, 1 mM DTT and 5% glycerol). The hTRPV4-RhoA complex was eluted by 5 CV of elution buffer (20 mM Tris pH 8.0, 150 mM NaCl, 0.03% LMNG, 0.003% CHS, 1 mM DTT, 5% glycerol, 0.150 mg mL⁻¹ FLAG peptide). The eluted protein complex was concentrated and further purified by size exclusion chromatography (SEC) on a Superose 6 Increase column (Cytiva Life Science) equilibrated with SEC buffer (20 mM Tris pH 8.0, 150 mM NaCl, 0.00075% LMNG, 0.000075% CHS, 0.0003% glycol-diosgenin (GDN; Anatrace), 1 mM DTT, and 5% glycerol).

### Cryo-EM sample preparation and data acquisition

TRPV4-RhoA peak fractions from SEC were concentrated to 0.8–1.2 mg mL⁻¹. All samples were incubated with different ligand conditions at 4 °C for 15–20 min prior to freezing grids. For the ligand-free-hTRPV4-RhoA sample, 2% (v/v) DMSO was added to the purified protein instead of ligands. For GSK279-TRPV4-RhoA, sample was incubated with 20 µM GSK2798745 (GSK279; MedChemExpress). For GSK101-TRPV4-RhoA, 20 µM GSK1016790A (GSK101; Tocris) and 2 mM Guanosine 5′-[γ-thio] triphosphate (GTPγS; Sigma-Aldrich) were incubated with the protein. For 4α-PDD-TRPV4-RhoA, sample was

incubated with 40 µM 4α-phorbol 12,13-didecanoate (4α-PDD; Sigma-Aldrich) and 2 mM GTPγS. All grids were prepared with a Leica EM GP2 plunge freezer (Leica) at 4 °C and 95% humidity. 3 µL of sample was applied to a freshly glow-discharged UltrAuFoil R1.2/1.3 300 mesh grid (Quantifoil), and blotted for 1.5–2.0 s, depending on the specific ligand conditions, to obtain optimal ice thickness for data collection.

Cryo-EM datasets for ligand-free-TRPV4-RhoA, GSK279-TRPV4-RhoA, GSK101-TRPV4-RhoA, and 4α-PDD-TRPV4-RhoA were collected with a Titan Krios microscope (FEI) operating at 300 keV equipped with a K3 detector (Gatan) with GIF BioQuantum energy filter (20 eV slit width: Gatan) in counting mode, using the Latitude-S automated data acquisition program (Gatan). Movie datasets were collected at a nominal magnification of 81,000x with a pixel size of 1.08 Å per pixel at specimen level. Each movie contained 60 frames over 4.6 s exposure time, using a dose rate of ~15 e⁻Å⁻²s⁻¹, resulting in the total dose of ~60 e⁻Å⁻². The nominal defocus ranged from −0.8 to −1.8 µm.

### Cryo-EM data processing

A total of 5666, 3877, 17,040, and 8226 movies were collected for the ligand-free-TRPV4-RhoA, GSK279-TRPV4-RhoA, GSK101-TRPV4-RhoA, and 4α-PDD-TRPV4-RhoA structures, respectively. All four datasets were processed similarly, as illustrated in Supplementary Fig. 1c with RELION 3.1[70] or 4.0[71] and cryoSPARC[72]. Beam-induced motion correction and dose-weighting were performed using MotionCor2[73], followed by CTF estimation using Gctf[74] in RELION. Micrographs were subsequentially selected based on astigmatism, CTF fit quality, CTF estimated maximum resolution, and defocus values. An initial set of particles were manually picked and subjected to a reference-free 2D classification ($k = 7$, $T = 2$), from which the best two to three classes were selected as reference for a templated-based auto-picking in RELION. Particles were re-centered and re-extracted, Fourier binned 4×4 (64-pixel box size), and imported to cryoSPARC. Two rounds of 2D classification with 50 classes were used to remove contamination and false noise picks, like chaperones and junk particles. The particles were imported to RELION and subjected to reference-free 2D classification ($k = 50$, $T = 2$), ignoring the CTFs until the first peak option. Classes were selected that showed apparent structure features of the TRPV4-RhoA complex. These particles were subsequently subjected to 3D classification ($k = 3$ or $4$, $T = 8$) with C1 symmetry with image alignment using a previously published TRPV1 map (EMD-23473, low-passed filtered to 60 Å) as a reference without masking. The class showing the apparent shape of the TRPV4-RhoA complex was selected, re-centered and re-extracted, and Fourier binned $2 \times 2$ for further classification. Refined particles at $2 \times 2$ Fourier binning were subjected to 3D classification with image alignment ($K = 3$, $T = 8$) and C1 symmetry. The class showing substantial densities for transmembrane helices and RhoA, was selected, re-centered and re-extracted without binning, and subjected to 3D classification without image alignment ($k = 2$ or $3$, $T = 8$–$12$) with a soft mask covering TRPV4 and RhoA with C4 symmetry imposed. The particles from the class with the best-resolved transmembrane helices were subjected to 3D auto-refinement with a soft TRPV4-RhoA mask. The refined particles were processed for CTF refinement[70], Bayesian polishing[75] then subjected to particle subtraction to resolve the strong density at the transmembrane domains (TMDs), followed by focused 3D classification. The tight mask for signal subtraction was made by subtracting out all unnecessary signals from the previous consensus 3D reconstruction, including detergent micelle, parts of the cytosolic domains, and RhoA. The subtracted particles by the tight mask were subjected to focused-3D classification without image alignment ($k = 2$, $T = 16$ or $20$). Particles comprising the best-featured class at TMDs were reverted to original particles, which were input to 3D auto-refinement with a TRPV4-RhoA full mask. Additional CTF refinement and Bayesian polishing were performed to improve the overall map quality. Finally, particles yielding the best 3D reconstruction from RELION were imported into cyroSPARC and

subjected to non-uniform (NU) refinement[76] with a full mask. To improve the EM density quality at the TMDs of TRPV4 the particles were subjected to particle subtraction of the four RhoA densities, followed by NU refinement. In order to resolve the unambiguous EM density around RhoA for GSK279-TRPV4-RhoA and GSK101-TRPV4-RhoA structures, we performed local refinement at the level of one monomeric ARD and RhoA by subtracting signals of the tetrameric TMDs, the other three ARDs, and RhoAs. A detailed data processing flowchart for the GSK101-TRPV4-RhoA structure is illustrated in Supplementary Fig. 3.

## Model building, refinement, and validation
For manual model building in Coot[77], the published cryo-EM structure of *Rattus norvegicus* TRPV1 (PDB: 7LP9) and published crystal structure of GDP-bound RhoA (PDB: 1FTN) were docked into the cryo-EM map for GSK279-TRPV4-RhoA. The angle between the ARD and TMD was first rigid-body adjusted into the EM densities, separately. Secondary structures were then rigid-body fit into the EM densities using bulky aromatic residues to ensure correct register assignment.

The focus refined TRPV4 channel map and ARD-RhoA focused map facilitated register assignment at S4-S5, S6-TRP flexible linkers and RhoA (Supplementary Fig. 4). The placement of individual residues was adjusted by rigid-body fitting then manually refined using real space refinement in Coot, with ideal geometric and secondary structure restraints. The GSK279-TRPV4-RhoA models served as the initial reference for model building of the GSK101-TRPV4-RhoA, ligand-free-TRPV4-RhoA, and 4α-PDD-TRPV4-RhoA structures. The restraints for ligands and lipid, including GSK279, GSK101, 4α-PDD, and CHS were generated from isomeric SMILES strings using the eLBOW tool[78] in PHENIX to fix bond lengths and angles. The manually built structure models with ligands were subjected to real-space refinement in PHENIX using cryo-EM maps with global minimization, rigid-body refinement and B-factor refinement[79].

Problematic regions identified by the MolProbity server (http://molprobity.biochem.duke.edu)[80], including geometry outliers and Ramachandran outliers, were manually adjusted in Coot. The FSC curves for the model against the full map and both half-maps were generated by comprehensive validation[81] in PHENIX. The FSCs were in good agreement with each other, indicating models were not over-fitted and refined. Structural illustrations and analysis were performed in Coot[77], PyMOL[82], UCSF Chimera[83], and UCSF ChimeraX[84]. For the figure preparations of cryo-EM density maps, UCSF ChimeraX and PyMOL were used.

## All-atom MD simulations
Three replicates for each of eight simulation assemblies were made in a mixed membrane of POPC:POPE:Cholesterol = 2:1:1. The eight simulation assemblies are 1) GSK279-TRPV4 in GSK279 pose I, 2) GSK279-TRPV4 in GSK279 pose II, 3) GSK279-TRPV4-RhoA-GDP in GSK279 pose I, 4) GSK101-TRPV4, 5) GSK101-TRPV4-RhoA-GTP, 6) 4α-PDD-TRPV4 in 4α-PDD pose I, 7) 4α-PDD-TRPV4 in 4α-PDD pose II, 8) 4α-PDD-TRPV4 in 4α-PDD pose III. The simulations were performed using the CHARMM36m force field (lipid, protein, nucleic acid[85–89]), TIP3P water model[90], CGenFF[91] (GSK279, GSK101, 4α-PDD). The initial simulation systems were assembled in CHARMM-GUI *Membrane Builder*[92–95] and equilibrated following the standard CHARMM-GUI six-step procedure. Additional equilibration of 50 ns was performed by gradually loosening protein backbone restraints from 50 to 0 kJ mol$^{-1}$ nm$^{-2}$. Hydrogen mass repartition was applied to the simulation systems[96] and a 4-fs time step was used for equilibration and production in OpenMM[97]. The van der Waals interactions were cut off at 12 Å with a force-switching function between 10 and 12 Å. Each system was held at a constant particle number, 1 bar pressure, and 300.15 K temperature (NPT). Most of the simulations were simulated for 800 ns, except for assemblies *2* (200 ns), and *6–8* (200 ns), to quickly examine ligand stability.

For ligand RMSD (root-mean-square-deviation) analysis of GSK101, GSK279, and 4α-PDD each individual VSLD site was aligned based on the helical residues (S1: 461–491, S2: 505–534, S3: 545–568, S4: 572–595, TRP: 729–748). Ligand RMSD was then calculated based on heavy atoms (for GSK101 and GSK279) or 4α-phorbol head group (for 4α-PDD). The distance between RhoA and TRPV4 ARD was calculated based on the centers of mass of the protein domains. RMSF (root-mean-square-fluctuations) of TRPV4 ARD were calculated based on alignment of the whole TRPV4 protein using the last 300 ns with 2 frames sampled from each ns. It was then averaged across the 12 replicates (3 replicas x tetramer) to derive the mean value and standard deviation. For ligand binding conformational ensemble, snapshots from each subunit were extracted at the end of simulations and aligned based on the VSLD cavity residues.

## Two-electrode voltage-clamp electrophysiology
The WT human TRPV4 DNA was subcloned into the pGEM-HE vector, the construct was linearized with NheI, and complementary RNA (cRNA) was synthesized by in vitro transcription using T7 RNA polymerase (Thermo Fisher). Defolliculated oocytes (Ecocyte, Austin, TX) were injected with cRNA for each of the constructs and incubated at 17 °C for 1–3 days in a solution containing 96 mM NaCl, 2 mM KCl, 1 mM MgCl$_2$, 1.8 mM CaCl$_2$, 5 mM HEPES, pH 7.6 (with NaOH), and gentamicin. For the two-electrode voltage-clamp recordings, oocyte membrane voltage was controlled using an OC-725C oocyte clamp (Warner Instruments). Data were filtered at 1–3 kHz and digitized at 20 kHz using pClamp software (Molecular Devices) and a Digidata 1440 A digitizer (Axon Instruments). Microelectrode resistances were 0.1–1 MΩ when filled with 3 M KCl. The external recording solution contained 100 mM KCl, 2 mM MgCl$_2$, 10 mM HEPES, pH 7.6 (with KOH). Agonists, antagonist, and ruthenium red were applied using a gravity-fed perfusion system that can exchange the 150 μL recording chamber volume within a few seconds.

## Patch clamp electrophysiology
After 1 day co-expression of GFP and wild type TRPV4 and its mutants, the whole-cell configuration patch clamp recordings were done at room temperature (22 ˚C). Data were acquired with an Axopatch 200B amplifier (Molecular Devices), currents were low-pass filtered at 2 kHz (Axopatch 200B) and digitally sampled at 5–10 kHz (Digidata 1440 A). Pipettes were pulled from borosilicate glass (1.5 mm O.D. x 0.86 mm I.D. x 75 mm L; Harvard Apparatus) using a Sutter P-97 puller and heat-polished to final resistances between 2 and 3 MΩ. 90% series resistance (Rs) compensation was used in all whole-cell recordings. Electrodes were filled with an intracellular solution containing 140 mM NaCl, 1 mM MgCl$_2$, 10 mM HEPES, and adjusted to pH 7.4 (NaOH). The extracellular solution contained 140 mM NaCl, 10 mM HEPES, 5 mM EGTA, pH 7.4 (NaOH), GSK101 and ruthenium red (RR) were applied using a gravity-fed perfusion system. Currents were recorded using a voltage ramp protocol consisting of 50 ms at a holding potential of −60 mV, 1000 ms ramp to +60 mV, followed by another 50 ms at 60 mV. All electrophysiological data analysis was done using Igor Pro 6.34 A (Wavemetrics).

## Antibodies and reagents
Primary antibodies used were rabbit anti-Myc (Cell Signaling Technology, 2272, used at 1:1000 for western blots), mouse anti-Myc (Cell Signaling Technology, 2276, used at 5 μg/ml for co-immunoprecipitation), rabbit anti-FLAG (Cell Signaling Technology, 2368, used for co-immunoprecipitation experiments involving C3 transferase at 1:1000), rabbit anti-GFP (Cell Signaling Technology, 2555, used for co-immunoprecipitation experiments involving C3 transferase at 1:1000), rabbit anti-GFP (Thermo Fisher Scientific, A-11122, used at 1:1000 for western blot), phospho-ERK1/2 (Cell Signaling Technology, 9101, used at 1:1000 for western blot), rabbit anti-RhoA (Cell Signaling

Technology, 2117, used at 1:1000 for western blot), rabbit anti-RhoB (Cell Signaling Technology, 2098, used at 1:1000 for western blot), rabbit anti-RhoC (Cell Signaling Technology, 3430, used at 1:1000 for western blot) Secondary antibodies used were HRP-conjugated monoclonal mouse anti-rabbit IgG, light chain specific (Jackson ImmuonoResearch, 211-032-171, clone 5A6-1D10, used at 1:100,000) and goat anti-rabbit IgG (Li-COR, 926-32211, used at 1:50,000).

## Co-immunoprecipitation
HEK293T cells were cultured in Dulbecco's Modified Eagle's Medium (DMEM) supplemented with 10% (v/v) fetal calf serum (FCS) and penicillin/streptomycin at 37 °C with 6% $CO_2$. Cells were transfected with Lipofectamine LTX with Plus Reagent (Thermo Fisher Scientific) and lysed 24 h after transfection in IP Lysis Buffer (Pierce, 25 mM Tris-HCl pH 7.4, 150 mM NaCl, 10 mM $MgCl_2$, 1% NP-40, 1 mM EDTA, 5% glycerol) supplemented with EDTA-free Halt protease inhibitor cocktail (Thermo Fisher Scientific). Cells were lysed for 15 min followed by centrifugation at 21,130 xg for 10 min. Supernatants were incubated with primary antibody bound to magnetic Protein G Dynabeads (Thermo Fisher Scientific) for 1 h at 4 °C followed by several washes in IP wash buffer (PBS, 0.2% Tween 20). To elute bound proteins, Laemmli sample buffer with β-mercaptoethanol was added to the beads and samples were heated for 10 min at 70 °C. Protein lysates were resolved on 4–15% TGX gels (Bio-Rad Laboratories) and transferred to PVDF membranes. Membranes were developed using SuperSignal West Femto Maximum Sensitivity Substrate (Thermo Fisher Scientific) and imaged using an ImageQuant LAS 4000 system (GE Healthcare).

## Calcium imaging
MN-1 cells were transfected with TRPV4-GFP (WT or mutant) constructs or co-transfected with TRPV4-GFP and RhoA-Myc (WT or mutant) constructs using Lipofectamine LTX. Calcium imaging was performed on a Zeiss Axio Observer. Z1 inverted microscope equipped with a Lambda DG-4 (Sutter Instrument Company, Novato, CA) wavelength switcher. Cells were bath-loaded with Fura-2 AM (8 μM, Life Technologies) for 45–60 min at 37 °C in calcium-imaging buffer (150 mM NaCl, 5 mM KCl, 1 mM $MgCl_2$, 2 mM $CaCl_2$, 10 mM glucose, 10 mM HEPES, pH 7.4). For hypotonic saline treatment, one volumes of NaCl-free calcium-imaging buffer was added to one volume of standard calcium-imaging buffer for a final NaCl concentration of 70 mM. Cells were imaged every 10 s for 30 s prior to stimulation with hypotonic saline or GSK101, and then imaged every 10 s for an additional 2 min. Calcium levels at each time point were computed by determining the ratio of Fura-2 AM emission at 340 nM divided by the emission at 380 nM. Data are expressed as Fura ratio over time.

## Reporting summary
Further information on research design is available in the Nature Portfolio Reporting Summary linked to this article.

## Data availability
Coordinates have been deposited in the Protein Data Bank with the PDB IDs − 8FC9 (human TRPV4-RhoA, ligand-free), 8FC7 (human TRPV4-RhoA, GSK279-bound closed), 8FCB (human TRPV4-RhoA, GSK101-bound, open), 8FC8 (human TRPV4 only, GSK101-bound, open), 8FCA (human TRPV4, 4α-PDD-bound, putative open) respectively. The cryo-EM maps have been deposited in the Electron Microscopy Data Bank with the IDs EMD-28977 (human TRPV4-RhoA, ligand-free), EMD-28975 (human TRPV4-RhoA, GSK279-bound closed), EMD-29030 (human TRPV4-RhoA, GSK279-bound closed, TRPV4-focused), EMD-29031 (human TRPV4-RhoA, GSK279-bound closed, ARD-RhoA focused), EMD-28976 (human TRPV4-RhoA and TRPV4 only, GSK101 bound, open), EMD-29331 (human TRPV4-RhoA and TRPV4 only,

GSK101 bound, open, TRPV4-focused), EMD-29332 (human TRPV4-RhoA and TRPV4 only, GSK101 bound, open, ARD-RhoA focused), EMD-28978 (human TRPV4, 4α-PDD-bound, putative open), respectively. The MD simulation data generated in this study have been deposited in the Zenodo OpenAIRE database under accession code 7996190. We have used the following published structures for the initial model building: 7LP9. Source data are provided with this paper.

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

## Acknowledgements

Cryo-EM data were screened and collected at the Duke University Shared Materials Instrumentation Facility (SMIF) and at the Pacific Northwest Center for Cryo-EM (PNCC) at OHSU. We thank Nilakshee Bhattacharya at SMIF and Janette Myers at PNCC for assistance with the microscope operation. We thank Ying Yin and Justin Fedor for critical manuscript reading and advice throughout the project and Yang Suo for assistance in data collection. This research was supported by National Institutes of Health grants R35NS097241 (S.-Y.L.), R35NS122306 (C.J.S), R01GM138472 (W.I.), K08NS102509 (B.A.M.) and Muscular Dystrophy Association 629305 (C.J.S.). A portion of this research was supported by NIH grant U24GM129547 and performed at the PNCC at OHSU and accessed through EMSL (grid.436923.9), a DOE Office of Science User Facility sponsored by the Office of Biological and Environmental Research. DUKE SMIF is affiliated with the North Carolina Research Triangle Nanotechnology Network, which is in part supported by the NSF (ECCS-2025064).

## Author contributions

D.K. conducted biochemical preparation, sample freezing, cryo-EM data collection, and processing, F.Z. conducted biochemical preparation and electrophysiology experiments, all under the guidance of S.-Y.L. D.K. and S.-Y.L. performed model building and refinement. B.A.M., and M.K. carried out immunoprecipitation and calcium imaging experiments under the guidance of C.J.S. S.F. conducted MD simulations under the guidance of W.I. S.-Y.L. D.K. C.J.S. B.A.M. J.M.S. and F.Z. wrote the paper.

## Competing interests

The authors declare no competing interests.
