## [Peer Review File · Nature Communications]

TRPV4-Rho GTPase complex structures reveal mechanisms of gating and diseaseREVIEWER COMMENTS

Reviewer #1 (Remarks to the Author):

"Structural insights into TRPV4-Rho GTPase signaling complex function and disease" by Kwon, Zhang-et-al from Seok-Yong Lee's group at Duke University, in collaboration with well-established Johns Hopkins investigators around Charlotte Sumner, reports very important new insights on cryo-EM structure of TRPV4 and molecular mechanisms of TRPV4-Rho GTPase signaling mechanisms.

These insights have direct relevance for understanding of TRPV4 neuro-channelopathy hereditary diseases, and also elucidate basic mechanisms of TRP ion channel function based on novel structural insights.

Several issues need to be addressed.

This is a general relevance paper.

Thus, there is a mandate to de-abbreviate

ARD

CD

VSLD

SF

and other "jargon" abbreviations not understood outside the silo of channel structuralists.

In the Abstract: "in the apo" - not suitable for a general relevance paper

Intro:

General knowledge re TRPV4 needs to be clarified, given that the paper is aiming at a general audience with all subdisciplines of life science represented.

TRPV4 is also activated by

- endogenous glycerophospholipids
- UVB
- mechanical stimulation other than shear

plus respective references.

TRPV4 also mediates

- pain (various isoforms)
- joint function
- barrier integrity for skin
- barrier integrity for vascular barriers in lung, BBB
- glial function for astrocytes and microglia
- neuronal function

plus respective references

Going through the ms.:

line 155-165

Please comment on postulated binding sites of endogenous TRPV4-activating lysolecithines, LPA and LPC, which both share a relevance of R746.

PMIDs 33819485, 36625071

line 286 nature, not Nature

line 297: this is a more complex issue: there are multiple instances where cell motility and metastatic capability are enhanced by TRPV4, yet clearly for tumor vascularization, critically involving TRPV4+ tumor vasculature endothelial cells, TRPV4 loss-of-function enhances thus worsens tumor/metastasis vascularization. Therefore, where inhibition of TRPV4 would on the one hand be beneficial for tumor cell invasion and metastasis, the opposite is the case for tumor/metastasis vascularization.

line 307-308. This is expressed in confusing manner

line 309-310. In that case, TRPV4 expression in an artificial membrane (= an acellular system) might produce an osmo- and/or shear sensitive system. Whereas such an experiment has not been reported (possibly because of non-publication bias against negative outcome studies), other studies suggest that TRPV4's responsiveness to physical cues, namely osmotic and mechanic stimulation, relies on additional

proteins that facilitate TRPV4's function in response to such cues. Possibly this could also extend to contribution of lipids to channel function, and in this respect specific setup of the directly peri-channel lipid bilayer. - Please discuss.

ref 45: PMID16571723 showed this 3 years earlier

line 320-321: re ligand-dependent gating - it should not be forgotten that potent and selective TRPV4-activator GSK101, as also used here, is lethal to all species it has ever been applied to, via induction of pulmonary alveolar edema. Therefore, the knowledge of a molecular and structurally-based mechanism of channel activation of GSK101 and related activators can only lead to a constructive insight, re rational design of therapeutics, what NOT to do. In this respect, a recent paper (PMID33819485) discussed mechanisms of TRPV4 channel activation by lysophosphatidylcholine, an endogenous glycerophospholipid which can be elevated in disease but obviously not causing pulmonary edema. LPC was found to have a suggested binding site C-term of residue R746 which was found relevant for LPC-mediated activation of TRPV4. Impact of this new insight for future rational design of TRPV4-activating therapeutics is discussed in this paper and can serve as guidance.

Reviewer #2 (Remarks to the Author):

In this manuscript, the authors determined the three-dimensional structures of human TRPV4 in complex with RhoA and three small molecules. They proposed that RhoA served as an auxiliary subunit of TRPV4, as the endogenous RhoA was co-purified with over-expressed TRPV4 and observed in cryo-EM structures. Based on these structures and other experiments, the authors gained important and unique insights regarding the gating mechanism of TRPV4. However, there are some concerns to be address:

Major concerns:

1. The identity of the extra density observed in complex with TRPV4. The authors claimed that this extra density near the ankyrin repeat domain corresponds to RhoA protein, however, more evidence is needed. For instance, as RhoA is one of the members of Rho-related GTP-binding proteins, it is very similar to RhoB and RhoC in both structure and primary sequence. In extended data fig. 1a, antibodies were used to detect RhoA, but no test of antibody specificity was performed. Moreover, in extended data fig. 6, the key residues for TRPV4 interaction are identical among RhoA, RhoB and RhoC. Therefore, the claim that this extra density underlies RhoA is not convincing, unless the authors could show that the HEK293 cells used to express TRPV4 does not endogenously express either RhoB or RhoC, or the specificity of the antibody used is so high that it detects only RhoA but not RhoB or RhoC.

To fully establish the extra density observed is the endogenous RhoA, the authors should first genetically knockout RhoA from the HEK293 cells and then use the RhoA-free cells to express TRPV4 protein. This extra density should then be absent. Then RhoA protein should be transfected and expressed in such RhoA-free HEK293 cells, this extra density should be observed again.

2. The quality of cryo-EM density maps. From Fig. 1d and Fig. 5a, it is difficult to clearly observe densities of sidechains in the maps, especially in the transmembrane domains. The authors admitted that “the EM density for the pore domain in the apo state and the 4a–PDD-bound structures was resolved sub-optimally” (line 106), however, though they suggested the map quality of GSK279 or GSK101 bound state was higher, in Fig. 1e, the maps of all four states exhibited similarly low quality at least in the S6 gate, where the sidechain of residues in S6 gate was not clearly observed.

Such an ambiguity in density maps, especially in the S6 gate, would largely hinder the interpretation of the state of TRPV4 channel. For instance, the pore radius profiles of GSK279 or GSK101 bound state calculated in Fig. 3d become questionable as the sidechain orientation would largely affect pore radius calculation. In fact, in the lower panels of Fig. 3b and 3c, it is clear that the sidechain of M680 and I715 does not fit in, which undermines any distance measurements. The authors should try to improve cryo-EM data quality, especially in critical regions like the pore of TRPV4.

3. The functional tests should be much improved. (a) the use TRPV4 DM is questionable. N456H and W737R mutations locate before S1 and in the C terminus, respectively, which are not far away from the ligand binding pocket of GSK molecules formed by the S1-S4 helixes. The authors tried to argue that “The putative 2-APB binding site in TRPV4 is located at a distance from the VSLD cavity” (line 143) is invalid, as the reference #30 they cited studied 2-APB binding in TRPV3 channel, not TRPV4. As we know, there are many binding pockets for 2-APB in TRPV channels, for instance, in TRPV2 channel, 2-APB binds to the vicinity of either S5 (Pumroy et al., NC 2022) or S4 and S4-S5 linker (Su et al., NChemB 2022), which are different from that in TRPV3. Therefore, the binding site of 2-APB in TRPV4 DM is still unknown, so how the binding of 2-APB would affect the binding of GSK molecules cannot be just ignored.

(b) the lack of concentration-response curves. To assess the effect of mutations in the GSK molecules binding pocket, the authors just arbitrarily used one concentration of GSK molecules and then compared the current amplitudes to one concentration of 2-APB activation, which is far from being solid. How would the point mutations affect the 2-APB activation? Though the authors have used a relatively high concentration of 5 mM 2-APB, a point mutation could introduce large changes in concentration-response curve so that 5 mM 2-APB no longer saturates the channel activation. The use of a single concentration of GSK molecules is even more problematic, as depending on the profile of concentration-response curve, in mutants the effect of a single concentration of either an agonist or antagonist could be either very large (where the concentration is near EC50 or IC50 value) or very small (where the concentration is at the foot or plateau of the curve). In this manuscript, no concentration-response

curve was measured for either GSK101 or GSK279, so that the current recordings in Fig. 2 cannot be directly interpreted as a proof of changes in binding of GSK molecules as the authors claimed.

(c) As the author can perform patch-clamp experiments to study interactions between GSK molecules and TRPV4 with point mutations, they should also do so for point mutations disrupting RhoA and TRPV4 interactions in addition to calcium imaging, as patch-clamp recordings is the gold standard in functional study of ion channels.

4. The interaction between RhoA mutants and TRPV4. In Fig. 4f and 4g left panels, the authors showed that mutating residues in RhoA that interact with TRPV4 would virtually abolish RhoA-TRPV4 interaction in Co-IP experiments, indicating these mutant RhoA cannot bind to TRPV4. However, also in Fig. 4f and 4g right panels, in calcium imaging where the HEK cells used still express the WT and endogenous RhoA, the additional expression of mutant RhoA should not compete and interfere with the WT RhoA inhibition of TRPV4, but what the authors observed was that the inhibition of TRPV4 was reduced! This cannot happen unless the WT and endogenous RhoA was removed with the expression of mutant RhoA. With the continuous presence of WT and endogenous RhoA, introducing any non-interacting RhoA mutation would not interfere the WT RhoA inhibition of TRPV4.

Other concerns:

1. In Fig. 2c, N474 critically interacts with GSK279, but this residue was not tested in patch-clamp recordings.

2. In line 260, Fig. 6 cannot be found. Should it refer to extended data fig. 6?

Reviewer #3 (Remarks to the Author):

TRPV4 mediates the Ca²⁺ influx and regulates many physiological processes including vascular tone, adipose thermogenesis, and inflammation. Kwon et al report structures of TRPV4-Rho complex in the apo and ligand-bound states and proposed ligand and Rho modulation mechanism of the TRPV4 channel activity. This is an interesting and timely study and would guide the development of new drugs.

Major concerns:

1. The recognition of ligands. The four structures were determined at 3.30 to 3.75 Å resolutions, which are relatively low for accurate assignment of the ligand molecules, as shown in both Fig. 1f and Fig. 2a.

The lowest resolution of the apo-state structure makes things even more complicated because it is difficult to judge whether the extra densities are from the ligand or introduced by the higher resolution of the ligand-bound structures. To make the structure data more convincing, I suggest the authors (i) improve the map quality for all four structures by collecting new data or re-processing data; (ii) perform MD to confirm the configuration of the three ligands; and (iii) clearly show the densities of not only the ligands but also their interacting residues in Fig. 2a, especially 4 α -PDD, which is not well separated from N474 side chain in Fig 1f.

2. The definition of the closed and open states of TRPV4. (i) The authors claim that among the four structures, the GSK101-bound is in the open states and the rest three are in closed states in Line 98-101 just based on the maps in Fig 1e, which is not convincing because at low resolutions density of residue sides chains are hardly observed in the maps. To me, I would say that only GSK279-bound are in closed states and the other three are in open states according to Fig. 1e. The authors should put the pore radius plots in Fig. 1 or move the definition of channel states backward. (ii) Please justify the rotamer of Ile715 in Fig. 3c in the open state and make corrections to the pore radii in Fig. 3d, as well as the definition of the open state of the GSK101-bound structure whenever necessary. Based on the map in Fig. 3c, the Ile715 rotamers should be adjusted. (iii) the GSK101-bound structure displays an enlarged gate but a narrowed filter, the latter of which is in contrast with the activation effect of GSK101 on TRPV4. Do the authors have any comments on this observation?

3. The RhoA modulation model. It is interesting to analyze the relationship between ligand activation and Rho density. I appreciate the focused 3D classification and the particle distribution analysis. Is the dissociation of RhoA induced by the binding of GSK101 or favoring the binding of GSK101? The structures and WB show that RhoA and TRPV1 are not in a 1:1 ratio and there are some TRPV4 channels in a non-RhoA-bound state before the binding of GSK101. If GSK101 binds mostly to the non-RhoA-bound TRPV4 and activates the channel, the model in Fig. 5f should be modified. To address this question, the authors may want to compare the ligand-binding sites in the high-resolution structures of both RhoA-bound and non-RhoA-bound TRPV4.

Minor comments:

1. Fig. 1d, label different maps

2. Line 176, "is too far to coordinate cations" should be changed to "is too far to directly coordinate cations".

Reviewer #4 (Remarks to the Author):

Structural insights into TRPV4-Rho GTPase signaling complex function and disease

Do Hoon Kwon, Feng Zhang, Brett A. McCray, Meha Kumar, Jeremy M. Sullivan, Charlotte J. Sumner & Seok-Yong Lee.

The Transient Receptor Potential Vanilloid 4 (TRPV4) is a polymodal ion channel involved in various processes, including osmoregulation, vascular control, and bone homeostasis. Gain-of-function (GOF) mutations in this channel result in neuromuscular disorders. A previous work showed that interactions of TRPV4 with the cytoskeleton remodeling GTPase RhoA are disrupted by these GOF mutations, resulting in enhanced channel activity. On the other hand, overexpression of RhoA reduces channel function. Kwon et al. provide the first structures of the full-length human TRPV4 channel, including open and close conformations with various ligands and in complex with RhoA. The human TRPV4-RhoA structure exhibits the canonical domain-swap tetrameric arrangement observed in other TRP channel structures. Importantly, this work provides insight into the structural bases of ligand-dependent TRPV4 gating. Moreover, the authors identified important contact sites between TRPV4 and RhoA, which overlap with the positions of mutations resulting in neuromuscular disorders. The main strength of this work is that the authors depicted the conformational changes during TRPV4 ligand-dependent gating and the interaction between the channel and RhoA. On the other hand, the data presented for the RhoA-dependent TRPV4 gating is less robust. I am enthusiastic about this work and consider it an important contribution to the field. Some issues need to be addressed to strengthen their conclusions.

Major critiques:

- 1) In Figure 1b, the authors should include the number of cells used to obtain the calcium imaging averages. The authors should add calcium imaging data of the TRPV4/RhoA upon activation with GSK101. Including patch-clamp electrophysiology or TEVC of the effect of RhoA on TRPV4 function will benefit the manuscript.
- 2) The authors should evaluate TRPV4 function using patch-clamp electrophysiology or TEVC in the absence or presence of RhoA inhibitors.
- 3) Do the authors know whether RhoA affects TRPV4 membrane expression? This should be addressed experimentally in the manuscript.
- 4) I understand the rationale of using the double mutant N456H/W737R to activate TRPV4 with 2APB. However, why did the authors not use osmotic stimuli, as shown in other figures, to rule out potential allosteric effects from these mutations?
- 5) In Figures 2d and 2e, the authors should use the appropriate statistical test (e.g., ANOVA or Kruskal-Wallis) when comparing the effects of the various mutations.

- 6) The authors should consider adding a new main figure that includes Figures 2f and 2g, as well as Extended Data Figures 5b and 5c. Merging these figures together would clearly convey the conformational movements.
- 7) The experiments with Cd²⁺-dependent blocking of TRPV4 nicely support the conformational changes observed in the agonist-bound structure. Although clear, the author should include the proper statistical analysis in Figure 3f.
- 8) In Figures 4d-g, the authors should provide experimental replicas (more than one) and quantification for the western blots. Moreover, as mentioned above in critique #1, the authors should include the number of cells used to obtain the calcium imaging averages, as well as functional analyses of the mutants using patch-clamp recording or TEVC.
- 9) The authors claim that “the degree of suppression of TRPV4 activity correlated with the interaction strength between the RhoA mutants and TRPV4 (Fig. 4d) (lines 246-248)”. It is unclear to me, from the results in Figure 4d, how the authors can tie the interaction strength to function. The results of single amino acid substitutions at position D263 match with an all or none effect in function (Fig. 4d). Please clarify this conclusion based on Figure 4d.
- 10) In Figure 4f, it is not clear to me that there is any correlation between interaction strength and function. E54H function (which is closest to RhoA WT) does not correlate with the Co-IP results. I would expect a stronger band for E54H. Please clarify this result.
- 11) In Figure 5, it is unclear to me how the authors consider the conformational changes in the ARD as essential gating steps (Lines 275-277) rather than a consequence of agonist-dependent TRPV4 gating. Without the structures of TRPV4 in the absence of RhoA and in the presence or absence of GSK101, it is difficult to extract any RhoA-dependent TRPV4 gating. The authors should consider toning down their conclusions regarding RhoA-dependent TRPV4 gating. For instance, changing the title of RhoA-dependent TRPV4 gating to something like “Proposed model for RhoA-mediated TRPV4 inhibition”.
- 12) In Figure 5f (right panel), the authors depict that the open state breaks the interaction between TRPV4 and RhoA. The authors should consider performing Co-IP experiments in the presence or absence of GSK101. These results could validate their model.

Minor critiques:

- 1) Line 144 (manuscript) should be mutant N474A/DM instead of D743A/DM.
- 2) Line 146 (manuscript) should be Figure 2d instead of 2c.
- 3) Line 220 references Figure 6, instead this should be Extended Data Figure 6 and Extended Data Figure 8.
- 4) In line 228, the authors mention the E50 mutation. However, this residue is not present in Figures 4b or 4c.
- 5) In lines 238-248, Figure 4d is repeatedly referenced, instead of other panels like 4e-g.

6) Although the sequence comparison is an ok predictor of the specific effect of TRPV4/RhoA, the authors could test the overexpression of RhoA on TRPV1 function. The manuscript could benefit from this experiment.

7) In line 255 to 258, the authors compare the surface area differences and cite Extended Data Figure 7g. Is there a better representation to highlight the surface area? It is not clear to me in the current figure.

We thank the reviewers for constructive criticism of our manuscript. We have performed many additional experiments to address reviewers' concerns. Our revision is extensive, and we believe that our revised manuscript is significantly improved thanks to the reviewers' suggestions.

REVIEWER COMMENTS

Reviewer #1 (Remarks to the Author):

"Structural insights into TRPV4-Rho GTPase signaling complex function and disease" by Kwon, Zhang-et-al from Seok-Yong Lee's group at Duke University, in collaboration with well-established Johns Hopkins investigators around Charlotte Sumner, reports very important new insights on cryo-EM structure of TRPV4 and molecular mechanisms of TRPV4-Rho GTPase signaling mechanisms. These insights have direct relevance for understanding of TRPV4 neuro-channelopathy hereditary diseases, and also elucidate basic mechanisms of TRP ion channel function based on novel structural insights.

Several issues need to be addressed.

This is a general relevance paper.

Thus, **there is a mandate to de-abbreviate**

ARD

CD

VSLD

SF

and other "jargon" abbreviations not understood outside the silo of channel structuralists.

In the Abstract: "in the apo" - not suitable for a general relevance paper → **Remove**

R) Thank you for your valuable feedback. We have made the appropriate modification and replaced "in the apo" with "in the ligand-free" in abstract.

Intro:

General knowledge re TRPV4 needs to be clarified, given that the paper is aiming at a general audience with all subdisciplines of life science represented.

TRPV4 is also activated by

- endogenous glycerophospholipids
 - UVB
 - mechanical stimulation other than shear
- plus respective references.

R) We've referred the papers and modified the text accordingly (PMID: 36625071, 23929777, 33537292)

TRPV4 also mediates

- pain (various isoforms)
- joint function
- barrier integrity for skin
- barrier integrity for vascular barriers in lung, BBB
- glial function for astrocytes and microglia
- neuronal function

plus respective references

R) We've referred the papers (PMID: 25281928, 25519495, 17008604)

Going through the ms.:

line 155-165

Please comment on postulated binding sites of endogenous TRPV4-activating lysolecithines, LPA and LPC, which both share a relevance of R746.

PMIDs 33819485, 36625071

R) We discussed this point in the discussion (see below)

line 286 nature, not Nature

R) We've changed it.

line 297: this is a more complex issue: there are multiple instances where cell motility and metastatic capability are enhanced by TRPV4, yet clearly for tumor vascularization, critically involving TRPV4+ tumor vasculature endothelial cells, TRPV4 loss-of-function enhances thus worsens tumor/metastasis vascularization. Therefore, where inhibition of TRPV4 would on the one hand be beneficial for tumor cell invasion and metastasis, the opposite is the case for tumor/metastasis vascularization.

R) Thank you for bringing up this interesting point. The goal of our paper is to present the concept of direct interaction between TRPV4 and RhoA, while the goal of this paragraph is to emphasize its involvement in cancer beyond the neuropathy. We do not intend to delve into the specific effects of TRPV4 activation/inhibition alongside RhoA on tumor in great details. Therefore, we replaced "the metastatic cascade and tumor vascularization" with "cancer". So the sentence reads as follows "Although these mutations may interrupt RhoA binding with several partners, these data are consistent with prior work demonstrating a role for TRPV4-Rho GTPase in cancer^{5,25,46}."

line 307-308. This is expressed in confusing manner

R) We clarified the sentence.

Line 372-373. "Osmosensors can sense either the changes in the extracellular water activity or the resulting changes in cell structure⁵⁴"

line 309-310. In that case, TRPV4 expression in an artificial membrane (= an acellular system) might produce an osmo- and/or shear sensitive system. Whereas such an experiment has not been reported (possibly because of non-publication bias against negative outcome studies), other studies suggest that TRPV4's responsiveness to physical cues, namely osmotic and mechanic stimulation, relies on additional proteins that facilitate TRPV4's function in response to such cues. Possibly this could also extend to contribution of lipids to channel function, and in this respect specific setup of the directly peri-channel lipid bilayer. - Please discuss.

R) Thank you. We included the possibility of other proteins and lipids in TRPV4's osmo-sensitivity.

ref 45: PMID16571723 showed this 3 years earlier

R) We replaced the reference.

line 320-321: re ligand-dependent gating - it should not be forgotten that potent and selective TRPV4-activator GSK101, as also used here, is lethal to all species it has ever been applied to, via induction of pulmonary alveolar edema. Therefore, the knowledge of a molecular and structurally-based mechanism of channel activation of GSK101 and related activators can only lead to a constructive insight, re rational design of therapeutics, what NOT to do. In this respect, a recent paper (PMID33819485) discussed mechanisms of TRPV4 channel activation by lysophosphatidylcholine, an endogenous glycerophospholipid which can be elevated in disease but obviously not causing pulmonary edema. LPC was found to have a suggested binding site C-term of residue R746 which was found relevant for LPC-mediated activation of TRPV4. Impact of this new insight for future rational design of TRPV4-activating therapeutics is discussed in this paper and can serve as guidance.

R) Thank you for this insightful point. We discussed this point in the discussion.

Line 385-389, Recent studies have revealed that lysophosphatidylcholine (LPC) acts as an endogenous agonist, binding to R746 within the TRP domain⁵⁵. It remains to be determined if this endogenous agonist employs an activation mechanism similar to that of synthetic agonists, which could potentially aid in further development of TRPV4 agonists.

Reviewer #2 (Remarks to the Author):

In this manuscript, the authors determined the three-dimensional structures of human TRPV4 in complex with RhoA and three small molecules. They proposed that RhoA served as an auxiliary subunit of TRPV4, as the endogenous RhoA was co-purified with over-expressed TRPV4 and observed in cryo-EM structures. Based on these structures and other experiments, the authors gained important and unique insights regarding the gating mechanism of TRPV4. However, there are some concerns to be address:

Major concerns:

1. The identity of the extra density observed in complex with TRPV4. The authors claimed that this extra density near the ankyrin repeat domain corresponds to RhoA protein, however, more evidence is needed. For instance, as RhoA is one of the members of Rho-related GTP-binding proteins, it is very similar to RhoB and RhoC in both structure and primary sequence. In extended data fig. 1a, antibodies were used to detect RhoA, but no test of antibody specificity was performed. Moreover, in extended data fig. 6, the key residues for TRPV4 interaction are identical among RhoA, RhoB and RhoC. Therefore, the claim that this extra density underlies RhoA is not convincing, unless the authors could show that the HEK293 cells used to express TRPV4 does not endogenously express either RhoB or RhoC, or the specificity of the antibody used is so high that it detects only RhoA but not RhoB or RhoC.

To fully establish the extra density observed is the endogenous RhoA, the authors should first genetically knockout RhoA from the HEK293 cells and then use the RhoA-free cells to express TRPV4 protein. This extra density should then be absent. Then RhoA protein should be transfected and expressed in such RhoA-free HEK293 cells, this extra density should be observed again.

R) Thank you for your valuable comments. We appreciate your concern regarding the specificity of the interaction of TRPV4 with RhoA. While we have used the RhoA-specific antibody (Cell Signaling Technology, RhoA (67B9) Rabbit mAb #2117) in our experiments, we acknowledge that it cannot rule out the possibility of TRPV4 interaction with the other Rho proteins. To address this concern, we conducted additional western blotting with the purified TRPV4 and Co-IP using monoclonal antibodies specific to RhoB and RhoC (Cell Signaling Technology, RhoB Antibody #2098 and RhoC (D40E4) Rabbit

mAb #3430) (see a new Extended Data Fig.1a and figures below). Our results confirm that not only RhoA but also RhoB and RhoC interact with TRPV4.

Extended Data Fig. 1a. SDS-PAGE and western blot analysis of purified hTRPV4-Rho GTPase.

Extended Data Fig. 2b. Close-up views of the $\alpha 6$ - $\beta 5$ region of RhoA, RhoB, and RhoC in TRPV4 ARD-Rho GTPase focused-map. The dotted circles indicate cryo-EM density unaccounted for in the respective models for RhoB and RhoC. Gray mesh indicates cryo-EM densities contoured at 0.15 thresholding.

However, when we inspect our final cryo-EM map, especially around $\alpha 6$ - $\beta 5$ region, the cryo-EM map is consistent with RhoA. This suggest that RhoA is the main RhoGTPase in the particles that were used for the final reconstruction. However, we cannot exclude the possibility that a significant portion of TRPV4 is bound to RhoB and RhoC. Therefore, we include the following sentences as well as the data to clarify this point.

Line99-102. "Although RhoA/B/C share a high degree of sequence homology, the focused cryo-EM map the docked RhoA model, and the sequence alignment suggests that the final cryo-EM reconstruction is more consistent with RhoA (Extended Data Fig. 2b and the below)

This suggests that RhoA may be the major Rho GTPase bound to TRPV4 in the final 3D reconstruction although we cannot exclude the possibility that a significant portion of TRPV4 is also bound to RhoB and RhoC. For this manuscript, we tentatively assign RhoA as the primary candidate based on the cryo-

EM map and previous identification of RhoA as a TRPV4 binding partner through an unbiased screen (McCray et al, 2021).”

The quality of cryo-EM density maps. From Fig. 1d and Fig. 5a, it is difficult to clearly observe densities of sidechains in the maps, especially in the transmembrane domains. The authors admitted that “the EM density for the pore domain in the apo state and the 4 α -PDD-bound structures was resolved sub-optimally” (line 106), however, though they suggested the map quality of GSK279 or GSK101 bound state was higher, in Fig. 1e, the maps of all four states exhibited similarly low quality at least in the S6 gate, where the sidechain of residues in S6 gate was not clearly observed.

Such an ambiguity in density maps, especially in the S6 gate, would largely hinder the interpretation of the state of TRPV4 channel. For instance, the pore radius profiles of GSK279 or GSK101 bound state calculated in Fig. 3d become questionable as the sidechain orientation would largely affect pore radius calculation. In fact, in the lower panels of Fig. 3b and 3c, it is clear that the sidechain of M680 and I715 does not fit in, which undermines any distance measurements. The authors should try to improve cryo-EM data quality, especially in critical regions like the pore of TRPV4.

R) We appreciate the opportunity to address your concerns and have revised our methodology and provided updated maps accordingly. Specifically, for the data of GSK279 and GSK101, we have generated TRPV4-focused maps using the subtracted Rho GTPase density to improve the quality of the transmembrane regions. Our improved maps and models based on the focused maps resulted in better defined S6 region of the models. They are provided in Fig. 1d, Fig. 4b, Fig. 4c, and figures below. Our analysis of S6 gate residues (M718 in GSK279 and I715 in GSK101) now provides clear and unambiguous results. Furthermore, we have made noticeable improvements in quality of the 4 α -PDD maps, which are included in the below figures and the attached new map. However, due to the intrinsic flexibility of the S6 helix in 4 α -PDD data, the register of the S6 helix is still sub-optimal. We have also attempted reprocessing the Apo data, but a better map was not obtained. For these reasons, we did not include the apo and 4 α -PDD data in our analysis of TRPV4 gating. We have updated Figure 1d.

GSK279 map quality in the transmembrane domain

GSK279 ligand density

GSK279
View I

GSK279
View II

GSK279
View III

GSK101 map quality in the transmembrane domain

GSK101 ligand density

GSK101
View I

GSK101
View II

GSK101
View III

Figs. 1e,f. (First row) Close-up view at the S6 gate of 3D reconstructions
 (Second row) Close-up view at the ligand binding site of 3D reconstructions

3. The functional tests should be much improved.

(a) the use TRPV4 DM is questionable. N456H and W737R mutations locate before S1 and in the C terminus, respectively, which are not far away from the ligand binding pocket of GSK molecules formed by the S1-S4 helices. The authors tried to argue that “The putative 2-APB binding site in TRPV4 is located at a distance from the VSLD cavity” (line 143) is invalid, as the reference #30 they cited studied 2-APB binding in TRPV3 channel, not TRPV4. As we know, there are many binding pockets for 2-APB in TRPV channels, for instance, in TRPV2 channel, 2-APB binds to the vicinity of either S5 (Pumroy et al., NC 2022) or S4 and S4-S5 linker (Su et al., NChemB 2022), which are different from that in TRPV3. Therefore, the binding site of 2-APB in TRPV4 DM is still unknown, so how the binding of 2-APB would affect the binding of GSK molecules cannot be just ignored.

R) We thank the reviewer for bringing up this point. First, we realized that we did not make clear our justification to use this double mutant. Patapoutian’s group previously identified that the 2-APB binding site involved two residues (N456 and W737) between the TRP domain and the intracellular region of TRPV3. Once they introduce these mutations into TRPV4, they saw that 2-APB could activate TRPV4. This study was referenced in our original manuscript (Hu et al, 2009). Our group then solved the structure of 2-APB bound TRPV3, confirming that the site is away from the VSLD (Zubcevic et al, 2019). Therefore, although there is no direct structural evidence of a 2-APB binding site in the TRPV4^{DM}, it is reasonable to assume that 2-APB binds to the analogous site that was observed in TRPV3. We clarified this point in our revision.

Line 159-172. “The shared binding region for GSK279 and GSK101 complicates mutagenesis studies to specifically examine GSK279 interactions with TRPV4. Therefore, we utilized an approach to create a 2-aminoethoxydiphenyl borate (2-APB) agonist binding site in TRPV4 distinct from the VSLD cavity. Patapoutian and colleagues previously demonstrated the 2-APB binding site in TRPV3 and created an analogous 2-APB responsive site in TRPV4 by site directed mutagenesis (N456H/W737R, denoted as TRPV4^{DM}) that enabled TRPV4^{DM} activation by 2-APB³⁵ (Extended Data Fig. 6a). As the 2-APB binding site in TRPV3 is located at a distance from the VSLD cavity³⁶, we predicted that 2-APB binding to TRPV4^{DM} would not interfere with either GSK279 or GSK101 binding. To verify that TRPV4^{DM} does not disrupt TRPV4 ion channel function or GSK101 binding, we demonstrated that 1) TRPV4^{DM} can be activated by either osmotic stimuli or GSK101 to a similar extent as the wild type TRPV4 (Extended

Data Figs. 6b,c) and 2) GSK101 binding site mutants Y553A, D743A, and F524A introduced onto the background of TRPV4^{DM} suppressed TRPV4 activation by GSK101 relative to that by 2-APB (Extended Data Fig. 6d), similar to the results from wild type TRPV4 background.”

Notably, to test the agonist GSK101 binding site, we conducted a dose-response experiment on TRPV4 wild type, not TRPV4^{DM}, and three other mutants (Y553A, N474A, F524A) in the wild type background (See Figs. 2d). Compared to the WT TRPV4, the dose-response curves of all three-point mutants were significantly right-shifted, with the Y553A mutation exhibiting an EC₅₀ approximately ~300 times higher than the WT TRPV4. These results are consistent with the results from our previous current ratio experiments using the TRPV4^{DM} construct. Also, upon request by reviewer 4, we performed calcium imaging experiments and found that TRPV4^{DM} does not disrupt GSK101 binding and shows retained osmosensitivity, which we included in our revision as well.

Fig. 2f. Mean normalized concentration-response relations for GSK101. Data are shown as mean ± S.E.M. (n = 3–5). The continuous curves are fits to the Hill equation with EC₅₀ and s (slope)

(b) the lack of concentration-response curves. To assess the effect of mutations in the GSK molecules binding pocket, the authors just arbitrarily used one concentration of GSK molecules and then compared the current amplitudes to one concentration of 2-APB activation, which is far from being solid. How would the point mutations affect the 2-APB activation? Though the authors have used a relatively high concentration of 5 mM 2-APB, a point mutation could introduce large changes in concentration-response curve so that 5 mM 2-APB no longer saturates the channel activation. The use of a single concentration of GSK molecules is even more problematic, as depending on the profile of concentration-response curve, in mutants the effect of a single concentration of either an agonist or antagonist could be either very large (where the concentration is near EC₅₀ or IC₅₀ value) or very small (where the concentration is at the foot or plateau of the curve). In this manuscript, no concentration-response curve was measured for either GSK101 or GSK279, so that the current recordings in Fig. 2 cannot be directly interpreted as a proof of changes in binding of GSK molecules as the authors claimed.

R) As mentioned above, regarding GSK101, we performed GSK101 dose-response curves of TRPV4 WT and mutants and found the results are consistent with the previous experiment using the TRPV4^{DM} and 2-APB.

Regarding GSK279 interaction analyses, we conducted dose-response experiments to determine the effect of 2-APB on TRPV4^{DM}, which revealed an EC₅₀ of ~ 300 μM (see below). For TRPV4 activation, we used 2 mM 2-APB, which is above the saturation concentration according to the concentration-response curve. Using 2mM 2-APB, we performed dose-response experiments on GSK279, with an IC₅₀ of ~350 nM. Both D546 and D743 mutants (in the background of the double mutant) do not express well, and thus the current amplitudes from cells expressing these mutants are small, making it challenging to conduct GSK279 dose-response experiments on these mutants despite our repeated trials for longer than six months. Nevertheless, we manage to carry out an additional experiment on D743A using 10 μM GSK279 to inhibit the currents induced by 2mM 2-APB, and we observed that the current could not be further inhibited compared to 4 μM GSK279 (Figure 2g). Combining this with our structural information, we suggest that these mutants affect GSK279's binding. We also performed all atom molecular dynamics simulation studies to confirm the binding poses of the ligands in this study (Fig. 2h-i).

Extended Data Figs. 6a,e.

(Left) TRPV4^{DM} mean normalized concentration-response relations for 2-APB. Data are shown as mean ± SEM (n = 4). The curves are fit to the Hill equation with EC₅₀ = 312 ± 12 μM, and s (slope) = 1.95.

(Right) Concentration-response curve for the effects of GSK279 on the 2 mM 2-APB-stimulated currents as a percent of the maximal inhibition response with the presence of 50 μM TRPV4 pore blocker ruthenium red. The values are expressed as a mean ± S.E.M. n=4-5 for each data point, IC₅₀=350.9 ± 38.1 nM, slope= -1.2

(c) As the author can perform patch-clamp experiments to study interactions between GSK molecules and TRPV4 with point mutations, they should also do so for point mutations disrupting RhoA and TRPV4 interactions in addition to calcium imaging, as patch-clamp recordings is the gold standard in functional study of ion channels.

R) Thank you for your comments. This study started from a collaboration amongst the Lee lab (structure and patch clamp electrophysiology) and the Sumner lab (cell biology, neurobiology, and calcium imaging). The Sumner lab has discovered the RhoA-TRPV4 interaction as well as TRPV4 neuropathy disease mutations and thus they have significant expertise in evaluating TRPV4 disease mutations. The disease mutations and RhoA-disrupting TRPV4 mutations are generally toxic to the cells, as they are gain-of-function mutations, and the Lee lab has not yet had success in in handling these mutants in patch clamp experiment. We agree that the patch clamp recording is the gold standard, but we made sure to carefully conduct calcium imaging, co-IP, and western blot experiments, and the conclusion from our studies is qualitative, simple, and clear (the disruption of the interaction interface between RhoA and TRPV4 increases TRPV4 activity). Further detailed, mechanistic studies utilizing patch clamp recordings would surely enhance our molecular level understanding of RhoA interaction with TRPV4, but such work will require extensive optimization followed by extensive

electrophysiological characterization and analyses, which will be an excellent topic for subsequent studies.

4. The interaction between RhoA mutants and TRPV4. In Fig. 4f and 4g left panels, the authors showed that mutating residues in RhoA that interact with TRPV4 would virtually abolish RhoA-TRPV4 interaction in Co-IP experiments, indicating these mutant RhoA cannot bind to TRPV4. However, also in Fig. 4f and 4g right panels, in calcium imaging where the HEK cells used still express the WT and endogenous RhoA, the additional expression of mutant RhoA should not compete and interfere with the WT RhoA inhibition of TRPV4, but what the authors observed was that the inhibition of TRPV4 was reduced! This cannot happen unless the WT and endogenous RhoA was removed with the expression of mutant RhoA. With the continuous presence of WT and endogenous RhoA, introducing any non-interacting RhoA mutation would not interfere the WT RhoA inhibition of TRPV4.

R) We acknowledge that the experiments presented in Figs. 5f-g were not sufficiently explained in the original submission. In the first set of experiments (Fig. 5f), TRPV4 mutants that fail to bind RhoA were over-expressed in MN-1 cells, and basal and stimulated calcium levels were noted to be increased. From this result, we inferred that interaction with endogenous RhoA was likely disrupted, leading to increased TRPV4 activity. However, to test this assumption more directly, we have performed additional experiments. We took advantage of the fact that the RhoA inhibitor C3 transferase binds to RhoA at the same site as TRPV4, suggesting that this inhibitor might be able to disrupt interaction with endogenous RhoA. Indeed, treatment of cells with C3 transferase markedly disrupted TRPV4-RhoA interaction by co-immunoprecipitation (Extended Data Figs. 9a-c, see (a-b) below). When MN-1 cells were treated with C3 transferase prior to calcium imaging, we found increased responses (c), similar to the effect of mutations that disrupt RhoA interaction. This result indicates that disruption of endogenous RhoA interaction is sufficient to cause gain of ion channel function and supports the conclusion that the increased channel activity shown in Fig. 5f is due to failure of TRPV4 mutants to interact with endogenous RhoA.

Extended Data Figs. 9a-c. **a**, Co-immunoprecipitation of HEK293T cells transfected with TRPV4-FLAG and RhoA-GFP with or without treatment with the RhoA inhibitor C3 transferase (0.5 $\mu\text{g/ml}$) for 12 hours demonstrates reduced TRPV4-RhoA interaction. **b**, Quantification of densitometry of TRPV4 bands on immunoblots, $n = 4$ (control) and 6 (C3). **c**, Averaged ratiometric calcium plots from ratiometric calcium imaging experiments. MN-1 cells were transfected with GFP-tagged TRPV4 plasmids and treated with C3 transferase (1 $\mu\text{g/ml}$) or vehicle for 2 hours and then stimulated with hypotonic saline. Baseline and hypotonic-stimulated calcium responses were then measured over time and then averaged, $n = 11$ wells per condition for control and 12 wells per condition for C3, with 20-40 transfected cells per well.

For the experiments in Fig. 5g, we over-expressed both TRPV4 and RhoA mutants. As we have shown previously, expression of RhoA in this paradigm markedly inhibits TRPV4 responses to both hypotonic saline (Fig. 5g) and GSK101 (McCray et al., 2021). This effect is presumably due to the inability of

endogenous RhoA to fully bind and inhibit over-expressed TRPV4, thus expression of additional RhoA results in observable inhibition of TRPV4. With these experiments, we found that RhoA mutants that fail to bind TRPV4 had reduced capacity to suppress TRPV4 channel function. This result provides further evidence that TRPV4-RhoA interactions exert potent inhibition on TRPV4-mediated calcium influx.

We have revised the text of this section to more clearly explain the rationale and methods for these experiments and the conclusions that we draw from them:

*“All mutants tested (R5E^{RhoA}, E54H/L/K^{RhoA}, D76A/L/K/R^{RhoA}, E183A/C/K^{TRPV4}, and D263A/L/K/N^{TRPV4}) substantially decreased the amount of immunoprecipitated partner proteins (TRPV4 and RhoA) (Fig. 5d). We then tested the effects of these mutations on TRPV4 function using ratiometric calcium imaging. With expression of TRPV4 mutants that fail to interact with RhoA (TRPV4 E183A/C/K or D263A/L/K/N), we found increased basal and hypotonic saline-induced calcium influxes, similar to the neuropathy mutant R269C mutation²². This suggests that disruption of interaction with endogenous RhoA leads to increased ion channel activity in the mutants. To directly test this possibility, we took advantage of the fact that the RhoA inhibitor exoenzyme C3 transferase of *C. botulinum* binds to RhoA within the TRPV4-RhoA interface. As predicted, treatment of cells with C3 transferase prior to co-immunoprecipitation strongly disrupted TRPV4-RhoA interaction (Extended Data Figs. 9a,b). In addition, treatment of MN-1 cells with C3 led to a marked increase in hypotonic saline-induced calcium influx (Extended Data Figs. 9c). These results suggest that disruption of RhoA interaction alone results in increased TRPV4 ion channel function. We then tested the effect of RhoA mutations on TRPV4 channel activity, in experiments in which we overexpressed both TRPV4 and RhoA. In this paradigm, we previously showed that over-expression of RhoA suppresses TRPV4 ion channel activity in response to hypotonic saline and GSK101 (McCray et al., 2021), perhaps due to the inability of endogenous RhoA to fully inhibit over-expressed TRPV4. Whereas expression of WT RhoA strongly suppressed both basal and hypotonic saline induced Ca²⁺ influx (Figs. 1b and 5d) consistent with prior results, RhoA mutants demonstrated reduced suppression of TRPV4 channel activity (Fig. 5d). Notably, there was a correlation between the degree of suppression of TRPV4 activity and the interaction strength between the RhoA mutants and TRPV4 (Fig. 5d), with the mutants with the highest residual TRPV4 binding (D76A/L and E54H) showing the strongest suppression of TRPV4 ion channel activity. These data demonstrate that TRPV4-RhoA interaction strength strongly correlates with TRPV4 channel activity, and that disruption of this interaction underlies the gain of function due to neuropathy mutations within the ARD.”*

Other concerns:

1. In Fig. 2c, N474 critically interacts with GSK279, but this residue was not tested in patch-clamp recordings.

R) Thank you for your comment. We've tested the N474A mutant with GSK279 (see below). Although there was a change in the basal activity of TRPV4, we didn't see any significant difference. Although N474 appear to interact with GSK279, the angle for the interactions do not seem optimal. Also, we cannot say about its energetic contribution based on the structure alone. We conclude that N474 is important for GSK101 binding to TRPV4, but not GSK279 binding to TRPV4.

2. In line 260, Fig. 6 cannot be found. Should it refer to extended data fig. 6?

R) We are sorry for the confusion. Yes, the reviewer is correct. It refers to Extended Data fig. 6. We fixed it.

Reviewer #3 (Remarks to the Author):

TRPV4 mediates the Ca²⁺ influx and regulates many physiological processes including vascular tone, adipose thermogenesis, and inflammation. Kwon et al report structures of TRPV4-Rho complex in the apo and ligand-bound states and proposed ligand and Rho modulation mechanism of the TRPV4 channel activity. This is an interesting and timely study and would guide the development of new drugs.

Major concerns:

1. The recognition of ligands. The four structures were determined at 3.30 to 3.75 Å resolutions, which are relatively low for accurate assignment of the ligand molecules, as shown in both Fig. 1f and Fig. 2a. The lowest resolution of the apo-state structure makes things even more complicated because it is difficult to judge whether the extra densities are from the ligand or introduced by the higher resolution of the ligand-bound structures. To make the structure data more convincing, I suggest the authors (i) improve the map quality for all four structures by collecting new data or re-processing data; (ii) perform MD to confirm the configuration of the three ligands; and (iii) clearly show the densities of not only the ligands but also their interacting residues in Fig. 2a, especially 4 α -PDD, which is not well separated from N474 side chain in Fig 1f.

R) Thank you for your suggestions. We have taken them into consideration and made the following improvements to our manuscript. Firstly, we have re-processed all the data to enhance the quality of the transmembrane domain region of TRPV4, as recommended. Specifically, we focused on TRPV4 using the subtracted Rho GTPase density and obtained new models based on the focused maps, which show improved qualities and fitting, as demonstrated in the updated figures (Fig. 1d, Fig. 4b, Fig. 4c, and the figures below). The densities of S6 gate residues (M718 in GSK279 and I715 in GSK101) now provide unambiguous analysis, and the ligand-binding regions are much clearer than before (Fig. 2a and the below figures). Secondly, as suggested, we have performed molecular dynamics (MD) simulations to validate the configurations of the three ligands, as shown in Figs. 2h,i and Extended Data Figs. 6i,j. The configurations of all three ligands in our current model are stable during the MD simulation time. For GSK279 and 4 α PDD we tested multiple binding poses to ensure that our binding pose is accurate. Finally, we have included the densities of ligand-interacting residues in Fig. 2a for clarity. We hope that these revisions will address your concerns.

GSK279 map quality in the transmembrane domain

GSK279 ligand density

GSK279
View I

GSK279
View II

GSK279
View III

GSK101 map quality in the transmembrane domain

GSK101 ligand density

GSK279
View I

GSK279
View II

GSK279
View III

Figs. 2h,i. **h**, Ligand-binding conformational ensemble from 12 sites (3 replicas x tetramer) at the end of 800-ns simulation of GSK101 (left), GSK279 pose I (middle) and GSK279 pose II (right). **i**, Ligand RMSD values of GSK101 show stable ligand binding with an average RMSD of 1.65 Å. Each trajectory represents a subunit (A/B/C/D) in one of the three replicas (left). Ligand RMSD values of GSK279 pose I show stable ligand binding with an average RMSD of 1.28 Å, except for one outlier ligand, rep2-D, which stomps out of the pocket (middle). Ligand RMSD values of GSK279 pose II show large deviations from the initial configuration with an average RMSD of 4.33 Å (right).

Extended Figs. 6i,j. **i**, Initial ligand-binding conformational poses of 4α-PDD pose I (left), 4α-PDD pose II (middle), and 4α-PDD pose III (right). **j**, Ligand RMSD values of 4α-PDD pose I show large deviations from the initial configuration (left). Ligand RMSD values of 4α-PDD pose II show large deviations from the initial configuration (middle). Ligand RMSD values of 4α-PDD pose III show stable ligand binding (right).

2. The definition of the closed and open states of TRPV4. (i) The authors claim that among the four structures, the GSK101-bound is in the open states and the rest three are in closed states In Line 98-101 just based on the maps in Fig 1e, which is not convincing because at low resolutions density of residue sides chains are hardly observed in the maps. To me, I would say that only GSK279-bound are in closed states and the other three are in open states according to Fig. 1e. The authors should put the pore radius plots in Fig. 1 or move the definition of channel states backward. (ii) Please justify the rotamer of Ile715 in Fig. 3c in the open state and make corrections to the pore radii in Fig. 3d, as well as the definition of the open state of the GSK101-bound structure whenever necessary. Based on the map in Fig. 3c, the Ile715 rotamers should be adjusted. (iii) the GSK101-bound structure displays an enlarged gate but a narrowed filter, the latter of which is in contrast with the activation effect of GSK101 on TRPV4. Do the authors have any comments on this observation?

R) Thank you for your comment. As mentioned in the text, we acknowledge the possibility that the apo and 4 α -PDD states may be open. However, the data quality for these states is sub-optimal even after reprocessing. Therefore, we have refrained from discussing the exact gating mechanism of these states given the less accurate data. Next, we have re-built our model based on the TRPV4 focused map, and among seven possible rotamer conformations for I715. We fitted both rotamer #1 and rotamer #2 reasonably as the cryo-EM of the side chain of this residue was not clear. However, based on better refinement statistics such as bond angle, bond length, and rotamer score, we have chosen rotamer #1. The degree of gate opening remains the same regardless of rotamer 1 or 2. We would also like to point out that this rotamer conformation is commonly observed in TRPV channels, as reported in previous studies (Kwon DH et al., 2021; Deng Z et al., 2020; Zubcevic L et al., 2019).

Regarding the enlarged gate and the narrowed filter for TRPV4 gating, we would like to note that we have recently observed similar noncanonical changes in the pore domain during TRPM8 channel gating in our published study (Yin Y et al., 2022). The exact reason why these changes occur in the pore domain is still unclear, but we speculate that the pore cavity of TRPV4 could potentially act as a sensor to detect changes in osmolality. We included mention of the TRPM8 study in the text.

3. The RhoA modulation model. It is interesting to analyze the relationship between ligand activation and Rho density. I appreciate the focused 3D classification and the particle distribution analysis. Is the dissociation of RhoA induced by the binding of GSK101 or favoring the binding of GSK101? The structures and WB show that RhoA and TRPV1 are not in a 1:1 ratio and there are some TRPV4 channels in a non-RhoA-bound state before the binding of GSK101. If GSK101 binds mostly to the non-RhoA-bound TRPV4 and activates the channel, the model in Fig. 5f should be modified. To address this question, the authors may want to compare the ligand-binding sites in the high-resolution structures of both RhoA-bound and non-RhoA-bound TRPV4.

R) Thank you for your valuable comment. Although WB suggest TRPV4: RhoA appears not 1:1, the final reconstruction suggested the ratio seems more stoichiometric with the GSK279 bound state. It is possible that RhoA-free TRPV4 particles are unstable so that they do not remain in the final 3D class.

The reviewer's question of whether GSK101 binding induces RhoA dissociation or GSK101 favors binding to RhoA-free TRPV4 is a good point, which is difficult to be resolved with current data. We speculate the former for the following reasons: 1) we added ligands to the apo TRPV4 before freezing and observed the stronger RhoA density in the GSK279-bound state and weaker RhoA density in the GSK101-bound state. 2) the final particle sizes (830k-890k) used for the analysis are similar between GSK101 and GSK279 states. 3) The non-RhoA-bound TRPV4 class, though it was resolved at a low resolution contains GSK101 similar to the RhoA-bound TRPV4 class (figure below).

We would also like to point out that we were not clear whether RhoA becomes more flexible or dissociates upon TRPV4 activation by GSK101, which we included as two possibilities in our first submission. We previously showed that osmotic stimuli activated TRPV4 dissociates RhoA (McCray et al, 2021). However, per reviewer 4's suggestion, we performed Co-IP experiments with and without GSK101 pre-treatment and found that GSK101 does not induce significant dissociation of RhoA from the TRPV4 complex. This data suggests that stimuli specific conformational changes of TRPV4 lead to differential effects on RhoA. We modified our model accordingly and discussed this point in the text and the discussion.

Cryo-EM densities of RhoGSK101 in RhoA-bound (a) and non-RhoA-bound (b) states at thresholding 0.34.

Minor comments:

1. Fig. 1d, label different maps

2. Line 176, "is too far to coordinate cations" should be changed to "is too far to directly coordinate cations".

R) Thank you for your comment. we have changed it (line 213).

Reviewer #4 (Remarks to the Author):

Structural insights into TRPV4-Rho GTPase signaling complex function and disease

Do Hoon Kwon, Feng Zhang, Brett A. McCray, Meha Kumar, Jeremy M. Sullivan, Charlotte J. Sumner & Seok-Yong Lee.

The Transient Receptor Potential Vanilloid 4 (TRPV4) is a polymodal ion channel involved in various processes, including osmoregulation, vascular control, and bone homeostasis. Gain-of-function (GOF) mutations in this channel result in neuromuscular disorders. A previous work showed that interactions of TRPV4 with the cytoskeleton remodeling GTPase RhoA are disrupted by these GOF mutations, resulting in enhanced channel activity. On the other hand, overexpression of RhoA reduces channel function. Kwon et al. provide the first structures of the full-length human TRPV4 channel, including open and close conformations with various ligands and in complex with RhoA. The human TRPV4-RhoA structure exhibits the canonical domain-swap tetrameric arrangement observed in other TRP channel structures. Importantly, this work provides insight into the structural bases of

ligand-dependent TRPV4 gating. Moreover, the authors identified important contact sites between TRPV4 and RhoA, which overlap with the positions of mutations resulting in neuromuscular disorders. The main strength of this work is that the authors depicted the conformational changes during TRPV4 ligand-dependent gating and the interaction between the channel and RhoA. On the other hand, the data presented for the RhoA-dependent TRPV4 gating is less robust. I am enthusiastic about this work and consider it an important contribution to the field. Some issues need to be addressed to strengthen their conclusions.

Major critiques:

1) In Figure 1b, the authors should include the number of cells used to obtain the calcium imaging averages. The authors should add calcium imaging data of the TRPV4/RhoA upon activation with GSK101. Including patch-clamp electrophysiology or TEVC of the effect of RhoA on TRPV4 function will benefit the manuscript.

R) We have now included details of the number of cells analyzed in the calcium imaging experiments. In our prior publication (McCray et al, 2021), we showed that expression of RhoA also suppresses calcium influx in response to GSK101 (see below).

From McCray et al., Supp Figure 4: **c.** MN-1 cells were transfected with WT TRPV4-GFP alone or in combination with WT RhoA followed by measurement of intracellular calcium levels in response to treatment with the TRPV4 agonist GSK101 (15 nM) at time = 0. Data represents an average of n = 11 independent coverslips per condition, each representing an average of 20-40 cells per coverslip. **d.** Average baseline and maximum Fura ratios in MN-1 cells from (c). Unpaired two-tailed t test, **p = 0.0040.

2) The authors should evaluate TRPV4 function using patch-clamp electrophysiology or TEVC in the absence or presence of RhoA inhibitors.

R) We took advantage of the fact that the RhoA inhibitor C3 transferase binds to RhoA at the same site as TRPV4, suggesting that this inhibitor might be able to disrupt interaction with endogenous RhoA. Indeed, treatment of cells with C3 transferase markedly disrupted TRPV4-RhoA interaction by co-immunoprecipitation (Extended Data Figs 9a,b and see below). When MN-1 cells were treated with C3 transferase prior to calcium imaging, we found increased responses (Extended Data Fig. 9c), similar to the effect of mutations that disrupt RhoA interaction. This result indicates that disruption of endogenous RhoA interaction is sufficient to cause gain of ion channel function.

Extended Data Figs. 9a-c. **a**, Co-immunoprecipitation of HEK293T cells transfected with TRPV4-FLAG and RhoA-GFP with or without treatment with the RhoA inhibitor C3 transferase (0.5 $\mu\text{g/ml}$) for 12 hours demonstrates reduced TRPV4-RhoA interaction. **b**, Quantification of densitometry of TRPV4 bands on immunoblots, $n = 4$ (control) and 6 (C3). **c**, Averaged ratiometric calcium plots from ratiometric calcium imaging experiments. MN-1 cells were transfected with GFP-tagged TRPV4 plasmids and treated with C3 transferase (1 $\mu\text{g/ml}$) or vehicle for 2 hours and then stimulated with hypotonic saline. Baseline and hypotonic-stimulated calcium responses were then measured over time and then averaged, $n = 11$ wells per condition for control and 12 wells per condition for C3, with 20-40 transfected cells per well.

This study is a collaboration between the Lee lab (structure and patch clamp electrophysiology) and the Sumner lab (cell biology, neurobiology, and calcium imaging). The Sumner lab has discovered the RhoA-TRPV4 interaction as well as TRPV4 neuropathy disease mutations and thus they have all the expertise handling TRPV4-RhoA interactions. We agree that the patch clamp recording is the gold standard, but we made sure to carefully conduct calcium imaging, co-IP, and western blot experiments, and the conclusion from our studies is clear (C3 inhibitor disrupt interface between RhoA and TRPV4 and thus increases TRPV4 activity).

3) Do the authors know whether RhoA affects TRPV4 membrane expression? This should be addressed experimentally in the manuscript.

R) We thank the reviewer for raising this important question. In our prior publication, we used surface biotinylation experiments to show that RhoA does not alter TRPV4 plasma membrane expression (McCray et al, 2021, Supp Fig. 4e, see below). This is now referenced in the manuscript as below:

“Overexpression of RhoA suppresses wild type (WT) TRPV4 channel-mediated calcium influx in cultured mouse motor neuron–neuroblastoma fusion (MN-1) cells in response to hypotonicity, demonstrating its ability to modulate TRPV4 function (Fig. 1a,b), and this effect occurs independent of changes in TRPV4 expression at the plasma membrane²².”

From McCray et al., Supp Figure 4: e. Surface biotinylation assay performed on HEK293T cells transfected with TRPV4-FLAG alone or in combination with RhoA-GFP showing that expression of RhoA does not affect the amount of TRPV4 expressed at the cell surface. Representative blot from two independent experiments.

4) I understand the rationale of using the double mutant N456H/W737R to activate TRPV4 with 2APB. However, why did the authors not use osmotic stimuli, as shown in other figures, to rule out potential allosteric effects from these mutations?

R) Thank you for your comment. To test whether introduction of the N456H and W737R mutations affected TRPV4 ion channel function, we transfected MN-1 cells with TRPV4-GFP N456H/W737R (TRPV4^{DM}) and performed calcium imaging experiments. Compared to TRPV4^{WT}, we found unchanged responses of TRPV4^{DM} to low-dose GSK101 stimulation. We also found preserved, albeit slightly reduced, responses to hypotonic stimulation in the TRPV4^{DM} mutant. These results demonstrate that introduction of N456H/W737R does not disrupt GSK101 binding and ion channel activity. This data is now included as Extended Data Figs. 5a,b (see below). We also did additional experiments of which results justify the use of the double mutant for our proposed experiment (Extended Data. Fig. 5).

Extended Data Figs. 6b,c. TRPV4^{DM} mutant shows preserved stimulated calcium influx responses. Averaged ratiometric calcium plots from ratiometric calcium imaging experiments. MN-1 cells were transfected with GFP-tagged TRPV4 plasmids and stimulated with (a) GSK101 (50 nM) or (b) hypotonic saline. Baseline and hypotonic-stimulated calcium responses were then measured over time and then averaged, n = 11 wells per condition, with 20-40 transfected cells per well.

5) In Figures 2d and 2e, the authors should use the appropriate statistical test (e.g., ANOVA or Kruskal-Wallis) when comparing the effects of the various mutations.

R) Thank you for your comments. Our intention in the previous Figures 2d and 2e was to compare the current ratios between TRPV4^{DM} (double mutant) and each point mutation in the background of TRPV4^{DM} individually, rather than comparing the current ratios amongst different mutants.

Fig. 2g. (Left) Summary of current inhibition by GSK279 relative to current from saturating 2-APB (2 mM) at room temperature. Values for individual oocytes are shown as open circles with mean \pm S.E.M. shown ($n = 3-9$). P values are calculated by two-tailed Student's t test.

Extended Data Fig. 6d. (Right) Probing the inhibitor binding site. TEVC recordings of hTRPV4^{DM} (N456H/W737R) and additional mutants made in the background of hTRPV4^{DM}, as indicated. Currents at -60 mV induced by 2 mM 2-APB then co-application of 4 μ M GSK279 followed by 20 μ M ruthenium red, as indicated by colored horizontal lines. Summary of current inhibition by 4 μ M GSK279 along with an additional 10 μ M GSK279 inhibition for D743A mutant relative to current from saturating 2-APB (2 mM) at room temperature.

In this case, we treat TRPV4^{DM} like a wild type and test the effect of introduced point mutation in the background of TRPV4^{DM}. Therefore, we are in an opinion that using a two-tailed Student's t-test is appropriate for our study because we simply want to test the effect of introduced point mutation on GSK279 binding compared to the reference construct. This approach has also been used in other literature, such as in the paper "Cytosine base editors induce off-target mutations and adverse phenotypic effects in transgenic mice" (Nature Communications, <https://doi.org/10.1038/s41467-023-37508-7>, Fig. 1c and Fig. 3a) and "Structural insights into TRPM8 inhibition and desensitization" (Science, <https://www.science.org/doi/10.1126/science.aax6672>, Fig. 3E, F). In both of these examples, multiple groups were compared, but the wild type was compared with each mutant individually.

6) The authors should consider adding a new main figure that includes Figures 2f and 2g, as well as Extended Data Figures 5b and 5c. Merging these figures together would clearly convey the conformational movements.

R) Thank you for your feedback. We have revised the figures as you suggested and incorporated them into Figure 3 and rearranged the figures accordingly. We agree that this figure arrangement better convey the ligand-dependent conformational arrangements.

7) The experiments with Cd²⁺-dependent blocking of TRPV4 nicely support the conformational changes observed in the agonist-bound structure. Although clear, the author should include the proper statistical analysis in Figure 3f.

R) Thank you for your comment, we added statistical analysis here using two tailed student t-test (Fig. 4f), since our intention is to compare how the mutants affects Cd²⁺'s blocking effect between wild type and each point mutants.

Fig. 4f. Representative time-course recording of hTRPV4 WT and mutants. Currents elicited by 5 μ M GSK101 and co-application with 10 μ M Cd²⁺ followed by 20 μ M ruthenium red (RR) as indicated by colored horizontal lines. The voltage was ramped from -60 mV to +60 mV in 300 ms every 2 seconds. The currents at -60 mV were used for the plot. Dotted blue lines indicate zero-current level. Right panel, summary of current inhibition by 10 μ M Cd²⁺ relative to 5 μ M GSK101 induced currents. Values for individual oocytes are shown as open circles ($n = 4-7$).

8) In Figures 4d-g, the authors should provide experimental replicas (more than one) and quantification for the western blots. Moreover, as mentioned above in critique #1, the authors should include the number of cells used to obtain the calcium imaging averages, as well as functional analyses of the mutants using patch-clamp recording or TEVC.

R) We appreciate this suggestion, and we have now performed all co-immunoprecipitation experiments in triplicate. We also now provide densitometry analysis of western blot band intensities as a method to quantify TRPV4-RhoA interactions. These data are now included as Extended Fig. 9d (see below).

We have also included more details of the number of cells analyzed in the calcium imaging experiments.

Extended Data Fig. 9d. Quantification of densitometry of TRPV4 bands on immunoblots, n = 3 for each mutant.

9) The authors claim that “the degree of suppression of TRPV4 activity correlated with the interaction strength between the RhoA mutants and TRPV4 (Fig. 4d) (lines 246-248)”. It is unclear to me, from the results in Figure 4d, how the authors can tie the interaction strength to function. The results of single amino acid substitutions at position D263 match with an all or none effect in function (Fig. 4d). Please clarify this conclusion based on Figure 4d.

R) Thank you for your comment. We apologize for any confusion in our previous statement. Based on the data presented in Fig. 5d, we observed that single amino acid substitutions at position D263 (D263A, D263L, and D263K) completely abolished the interaction between RhoA and TRPV4. Consequently, there were no significant differences in channel activity among these mutants. This result is consistent with the mutation of R5 in RhoA, which forms a salt bridge with D263 in TRPV4 directly. We toned down about our claim about the interaction strength and function and remove such claim in the abstract and the discussion.

10) In Figure 4f, it is not clear to me that there is any correlation between interaction strength and function. E54H function (which is closest to RhoA WT) does not correlate with the Co-IP results. I would expect a stronger band for E54H. Please clarify this result.

R) We agree that the correlation between TRPV4-RhoA interaction and functional effects on calcium influx with various TRPV4 and RhoA mutants was overstated. With this assertion, we are primarily referring to the data for the D76A/L and E54H RhoA mutants which show the highest degree of residual TRPV4 interaction by co-immunoprecipitation and the most preserved suppressive function on TRPV4 calcium influx. We have revised the text to more specifically highlight this data. The revised text is below:

“Notably, there was a correlation between the degree of suppression of TRPV4 activity and the interaction strength between the RhoA mutants and TRPV4 (Fig. 5d), with the mutants with the

highest residual TRPV4 binding (D76A/L and E54H) showing the strongest suppression of TRPV4 ion channel activity.”

As mentioned above, we removed claims about interaction strength and function from the abstract and the discussion.

11) In Figure 5, it is unclear to me how the authors consider the conformational changes in the ARD as essential gating steps (Lines 275-277) rather than a consequence of agonist-dependent TRPV4 gating. Without the structures of TRPV4 in the absence of RhoA and in the presence or absence of GSK101, it is difficult to extract any RhoA-dependent TRPV4 gating. The authors should consider toning down their conclusions regarding RhoA-dependent TRPV4 gating. For instance, changing the title of RhoA-dependent TRPV4 gating to something like “Proposed model for RhoA-mediated TRPV4 inhibition”.

R) Thank you for your comment. We toned down the conclusion, modified our model, and also changed the title as suggested (we did not include “proposed model” due to the length limit of the title).

12) In Figure 5f (right panel), the authors depict that the open state breaks the interaction between TRPV4 and RhoA. The authors should consider performing Co-IP experiments in the presence or absence of GSK101. These results could validate their model.

R) We thank the reviewer for bringing up this interesting and important point. To test how treatment with GSK101 affects TRPV4-RhoA interaction, we performed co-immunoprecipitation experiments in the presence of varying GSK101 doses and treatments durations. With these experiments, we did not find a significant dissociation of TRPV4-RhoA complexes in the presence of GSK101, consistent with some of our prior work (McCray et al., 2021, Supp Fig. 5g). In contrast, our previously published work showed that hypotonic stimulation does induce dissociation of TRPV4 and RhoA (McCray et al, 2021, Fig. 5i-j). These data, taken together, suggest that stimulus-specific conformational changes of TRPV4 lead to either RhoA release from or increased flexibility within the complex, which we find intriguing. We included the new data, modified our model, and discussed the result in the revision.

Extended Dat Fig. 9e. TRPV4-GFP and RhoA-Myc expressed in 293T cells, 1:2 ratio, IP: anti-Myc. Cells were treated with GSK101 prior to lysis, either 500 nM x10 minutes or 100nM x 12 hours. GSK101 (500 nM) was added to the lysis buffer and wash buffer for indicated samples.

Minor critiques:

1) Line 144 (manuscript) should be mutant N474A/DM instead of D743A/DM.

R) We've changed it.

2) Line 146 (manuscript) should be Figure 2d instead of 2c.

R) We've changed it.

3) Line 220 references Figure 6, instead this should be Extended Data Figure 6 and Extended Data Figure 8.

R) We've changed it.

4) In line 228, the authors mention the E50 mutation. However, this residue is not present in Figures 4b or 4c.

R) We've changed "E50" to "E54".

5) In lines 238-248, Figure 4d is repeatedly referenced, instead of other panels like 4e-g.

R) We've fixed it.

6) Although the sequence comparison is an ok predictor of the specific effect of TRPV4/RhoA, the authors could test the overexpression of RhoA on TRPV1 function. The manuscript could benefit from this experiment.

R) Thank you for your suggestion. We acknowledge that investigating the interaction between Rho GTPase and other members of the TRPV family would be valuable. However, we feel that such studies are beyond the scope of the current manuscript, which is focused on TRPV4. Furthermore, as a proper assessment would require evaluation of all TRPV members, not just TRPV1, such analysis would require substantial additional time and resources. Because of the extensive revision experiments we performed, including cryo-EM processing, model building, electrophysiology, co-IP, and molecular dynamics, we were unable to address this specific point within the resubmission deadline. However, we agree that such studies would be interesting and would be an excellent topic for subsequent studies. We hope that the reviewer agrees with us with these points.

7) In line 255 to 258, the authors compare the surface area differences and cite Extended Data Figure 7g. Is there a better representation to highlight the surface area? It is not clear to me in the current figure.

R) Thank you for your feedback. We have revised Extended Data Fig. 7g to emphasize the surface changes in ARD using surface views, but because the surface area difference is small (by 70 Å²), our figure does not convey the difference in surface areas as we had hoped.

Extended Data Fig. 7f. Comparison of the interaction interface between GSK279-hTRPV4-RhoA-GDP and GSK101-hTRPV4-RhoA-GTP γ S. Arrows indicate conformational changes.

REVIEWERS' COMMENTS

Reviewer #1 (Remarks to the Author):

The paper has been revised extensively and the authors are to be commended for an overall excellent job.

This is a novel, interesting and generally highly relevant contribution now.

Minor-minor modifications suggested are the Intro, which in its revised version is now more well-balanced for a general readership.

The following suggested references will round up the picture:

TRPV4 - skin barrier function PMIDs 17068482, 36210147

TRPV4 - BBB, blood CSF barrier PMIDs 25681460, 25914628

TRPV4 - glia PMIDs 17719182, 22331560

TRPV4 - CNS neurons PMID 17301165

Reviewer #3 (Remarks to the Author):

The authors have addressed all my concerns and I have no further question. I support the publication of this manuscript now.

Reviewer #4 (Remarks to the Author):

Overall, the authors have addressed my questions satisfactorily—no further comments.

Reviewers' Comments:

Reviewer #1 (Remarks to the Author):

The paper has been revised extensively and the authors are to be commended for an overall excellent job.

This is a novel, interesting and generally highly relevant contribution now.

Minor-minor modifications suggested are the Intro, which in its revised version is now more well-balanced for a general readership.

The following suggested references will round up the picture:

TRPV4 - skin barrier function PMIDs 17068482, 36210147

TRPV4 - BBB, blood CSF barrier PMIDs 25681460, 25914628

TRPV4 - glia PMIDs 17719182, 22331560

TRPV4 - CNS neurons PMID 17301165

R) We thank the reviewer #1's suggestions and have now included the references in the text.

Reviewer #2 (Remarks to the Author)

The authors have satisfactorily addressed all my concerns.

R) We thank the reviewer #2 for the constructive criticisms, which helped improve our manuscript significantly..

Reviewer #3 (Remarks to the Author):

The authors have addressed all my concerns and I have no further question. I support the publication of this manuscript now.

R) We thank the reviewer #3 for the constructive criticisms, which helped improve our manuscript significantly..

Reviewer #4 (Remarks to the Author):

Overall, the authors have addressed my questions satisfactorily—no further comments.

R) We thank the reviewer #4 for the constructive criticisms, which helped improve our manuscript significantly..